# Anticancer Effects of BRD4 Inhibitor in Epithelial Ovarian Cancer

**DOI:** 10.3390/cancers16050959

**Published:** 2024-02-27

**Authors:** Yeorae Kim, Wook-Ha Park, Dong-Hoon Suh, Kidong Kim, Jae-Hong No, Yong-Beom Kim

**Affiliations:** 1Department of Obstetrics and Gynecology, Seoul National University Bundang Hospital, 82 Gumi-ro, 173 Beon-gil, Bundang-gu, Seongnam 13620, Republic of Korea; delight005@snu.ac.kr (Y.K.); re_member18@naver.com (W.-H.P.); sdhwcj@naver.com (D.-H.S.); kidong.kim.md@snubh.org (K.K.); jhno@snu.ac.kr (J.-H.N.); 2Department of Obstetrics and Gynecology, Seoul National University College of Medicine, 103 Jongno-gu, Seoul 03080, Republic of Korea

**Keywords:** ovarian cancer, BRD4, anticancer effect

## Abstract

**Simple Summary:**

This study investigated a new drug, OPT-0139, an inhibitor of BRD4, a protein involved in ovarian cancer. OPT-0139 treatment reduced the growth and viability of ovarian cancer cells, induced cell death, and inhibited cancer cell division, suggesting its potential as a treatment agent. When combined with cisplatin, a chemotherapy drug, OPT-0139 was more effective in killing cancer cells and mitigating tumor growth. These findings suggest that OPT-0139, either alone or in combination with cisplatin, is a promising therapeutic agent for ovarian cancer.

**Abstract:**

Efforts have been made to develop bromodomain inhibitors as cancer treatments. Sub-pathways, particularly in ovarian cancer, affected by bromodomain-containing protein (BRD) remain unclear. This study verified the antitumor effects of a new drug that can overcome OPT-0139-chemoresistance to treat ovarian cancer. A mouse xenograft model of human ovarian cancer cells, SKOV3 and OVCAR3, was used in this study. Cell viability and proliferation were assessed using MTT and ATP assays. Cell cycle arrest and apoptosis were determined using flow cytometry. BRD4 and c-Myc expression and apoptosis-related molecules were detected using RT-PCR and real-time PCR and Western blot. We confirmed the OPT-0139 effect and mechanism of action in epithelial ovarian cancer. OPT-0139 significantly reduced cell viability and proliferation and induced apoptosis and cell cycle arrest. In the mouse xenograft model, significant changes in tumor growth, volume, weight, and BRD4-related gene expression were observed, suggesting the antitumor effects of BRD4 inhibitors. Combination therapy with cisplatin promoted apoptosis and suppressed tumor growth in vitro and in vivo. Our results suggest OPT-0139, a BRD4 inhibitor, as a promising anticancer drug for the treatment of ovarian cancer by inhibiting cell proliferation, decreasing cell viability, arresting cell cycle, and inducing apoptosis.

## 1. Introduction

Epithelial ovarian cancer (EOC) is the most aggressive gynecological cancer. It was reportedly the fifth leading cause of cancer-related deaths among women in the United States in 2018, with 21,410 newly diagnosed cases and 13,770 deaths by 2021 [1]. Furthermore, it is noteworthy that according to the Korea National Statistics, ovarian cancer has been identified as one of the top 10 leading causes of morbidity among women, with 3221 newly diagnosed cases reported in 2021. The crude incidence rate for this malignancy is reported as 12.5 per 100,000 individuals [2]. EOC is an aggressive malignancy; therefore, recurrence is common (almost 70–80%), and resistance to standard chemotherapy (platinum-based regimens) is frequent [3]. Few treatment options are available for patients with recurrent ovarian cancer after primary optimal debulking surgery and completion of front-line chemotherapy. Recurrence treatments include chemotherapy regimen changes and targeted therapies. Therefore, efforts have been made to overcome chemoresistance and decrease recurrence rates [3,4,5].

EOCs are characterized by high mortality and recurrence rates. This was the primary reason for developing this novel treatment option. One currently studied novel treatment option is targeted therapy, including poly ADP-ribose polymerase (PARP) inhibitor usage [6,7,8]. Promoting cell death by exploiting tumor DNA damage is the core mechanism of anticancer-targeted therapy. The epigenetic approach has recently gained attention, with bromodomain and extra-terminal domain (BET) as examples [9,10].

BET proteins are epigenetic readers consisting of four proteins: BRD2, BRD3, BRD4, and bromodomain testis-specific proteins (BRDT) [10]. BRD4 is overexpressed in various cancer cells, such as breast, colorectal, prostate, and gastric [11,12,13,14]. The small molecule BRD4 leads to dysregulated gene expression and promotion of cancer cell survival and proliferation. BRD4 inhibitors typically work by competitively binding to the bromodomain of BRD4, preventing its interaction with acetylated histones and other transcriptional co-factors [9,10]. A novel drug, OPT-0139, also acts as a competitive binding monovalent molecule. This disrupts the normal transcriptional machinery, leading to downregulating oncogenes and upregulating tumor-suppressor genes [9,10,15]. BRD4 inhibitors can also cause cell cycle arrest and induce apoptosis in cancer cells, further inhibiting their proliferation [9,10,15].

Concerning ovarian cancer, a comprehensive analysis of The Cancer Genome Atlas (TCGA) database revealed that 9% of ovarian cancer patients exhibited elevated BRD4 expression [16]. Despite efforts to use bromodomain inhibitors as anticancer drugs, few studies have investigated the correlation between BRD4 and ovarian cancer pathogenesis [17,18]. Recent studies mainly focus on the relationship between BRD4 inhibition and PARP inhibitors [18]. However, the correlation between ovarian cancer treatment and BRD4 inhibitors remains unclear due to the heterogeneous carcinogenic characteristics of ovarian cancer, differing from those of other cancers [19].

To address the challenges in treating ovarian cancer, our objective is to evaluate the antitumor effect of a novel drug, OPT-0139 (Appendix A), capable of overcoming the limitations of current ovarian cancer treatments. We hypothesized that a BRD4 inhibitor may have an anticancer effect either alone or in combination with cisplatin in preclinical settings with ovarian cancer. This study represents the initial exploration of the combination of a BRD4 inhibitor and the classic cytotoxic agent, cisplatin, in ovarian cancer cells.

## 2. Materials and Methods

### 2.1. Antibodies (Abs), Media and Reagents

Anti-BRD4 (ab244221; mouse Ab; 1:1000) and anti-BCL-2 (ab692; mouse Ab; 1:1000) Abs were purchased from Abcam (Cambridge, MA, USA). Anti-cleaved caspase-3 (9661; rabbit Ab; 1:1500), anti-cleaved PARP (5625; rabbit Ab; 1:1000), and β-actin (3700; mouse Ab; 1:4000) Abs were purchased from Cell Signaling Technology (CST, Danvers, MA, USA). RPMI-1640 (Roswell Park Memorial Institute-1640) culture media and fetal bovine serum (FBS) were purchased from Gibco (Thermo Fisher Scientific, Inc., Carlsbad, CA, USA). OPT-0139 was obtained from JBKLAB Co., Ltd. (Seongnam, Republic of Korea), and cisplatin was purchased from Sigma Aldrich (St. Louis, MO, USA) and dissolved in dimethyl sulfoxide (DMSO) for in vitro studies and 0.9% saline for animal studies.

### 2.2. Cell Culture

Human ovarian cancer cell lines (SKOV3 and OVCAR3) were purchased from the American Type Culture Collection (Manassas, VA, USA). An immortalized normal human ovarian surface epithelial cell line (HOSEpic) was obtained from ScienCell Research Laboratories, Inc. (Carlsbad, CA, USA). The cells were maintained in Roswell Park Memorial Institute-1640 culture (RPMI-1640; Gibco; Thermo Fisher Scientific, Inc., Carlsbad, CA, USA) supplemented with 10% (*v*/*v*) FBS, 100 mg/mL streptomycin, and 100 U/mL penicillin (P/S; Gibco; Thermo Fisher Scientific, Inc.) and incubated at 37 °C, 5% CO_2_.

### 2.3. Cell Viability Assay

Cell viability was assessed utilizing PrestoBlue Cell Viability Reagent (Invitrogen, Carlsbad, CA, USA). Cells were placed in 96-well plates with 100 µL of complete media (containing 10% FBS and 1% P/S-supplemented media) per well and incubated overnight at 95% humidity and 5% CO_2_. Following the overnight incubation, OPT-0139 was applied at concentrations ranging from 0 to 10 µM in 100 µL of complete media for a duration of 48 h. The final DMSO concentration in each well was maintained below 0.1%. Subsequently, cells were exposed to 10% PrestoBlue in the dark for 1 h. Absorbance, normalized to cell number (absorbance/cell number), was measured at 540 nm using a plate reader. Statistical analyses were conducted using GraphPad Prism version 9.4.1 for Windows, with significance set at *p* < 0.05, and higher significance at *p* < 0.01, in comparison to the control.

### 2.4. Cell Proliferation Assay

Cell proliferation was determined using Cell Counting Kit-8 (CCK-8) reagent (Dojindo Laboratories Co., Ltd., Tokyo, Japan). The cells were seeded into 96-well plates at 1 × 10^4^ cells/well density and treated with different OPT-0139 concentrations (0, 0.01, 0.1, 0.5, 1, 5, and 10 μM) for 48 h. The CCK-8 assay was performed 48 h after OPT-0139 treatment. The serum-free medium was replaced, and 10 μL of CCK-8 was added to each well. After incubation at 37 °C and 5% CO_2_ for 1 h, optical density was measured at 450 nm. Each measurement was performed in triplicate.

### 2.5. Immunofluorescence Staining

Immunofluorescence staining was performed by seeding cells at a density of 5 × 10^4^ cells/well on a 15 mm poly-L-lysine-coated cover glass in a 6-well plate, followed by a 24 h incubation at 37 °C. The cells underwent fixation with 4% paraformaldehyde, permeabilization with 0.1% Triton X-100 for 15 min at room temperature, and triple washing with phosphate-buffered saline (PBS). Subsequently, cells were blocked with 1% bovine serum albumin (Sigma Aldrich, St. Louis, MO, USA) for 60 min at room temperature. Following three 10 min washes with PBS, the samples were incubated overnight at 4 °C with an anti-BRD4 monoclonal antibody. After three 3 min PBS washes, cells were exposed to Alexa Fluor Plus 488 conjugated Goat anti-rabbit immunoglobulin G (1:1000, Thermo Fisher Scientific, Waltham, MA, USA) for 2 h at room temperature. Mitochondria were stained with MitoTracker Orange (1:5000, Thermo Fisher Scientific, Waltham, MA, USA) for 15 min at room temperature. Visualization and photography of fluorescence-labeled HOXB9 and mitochondria were conducted using a fluorescence microscope.

### 2.6. Caspase-3 Activity

Cells were initially plated at a density of 1 × 10^4^ cells/well in 96-well plates with white walls, followed by a 24 h incubation period. Subsequently, they were subjected to OPT-0139 treatment at concentrations of 0, 0.01, 0.1, 0.5, 1, 5, and 10 μM for a duration of 48 h. The treated cells were then incubated with 100 μL of Caspase-Glo 3/7 Reagent (Promega, Madison, WI, USA) at room temperature for 30 min. Luminescence readings for each sample were obtained using a luminometer (Molecular Devices, San Jose, CA, USA). All data were presented as percentages relative to the control.

### 2.7. Annexin V FITC and Propidium Iodide (PI) Apoptosis Detection Assay

Annexin V and propidium iodide staining were conducted employing an annexin V FITC/PI apoptosis detection assay kit (Invitrogen, Carlsbad, CA, USA). Cells were initially plated in 6-well dishes at a density of 5 × 10^5^ cells/well. OPT-0139 was introduced at concentrations of 0, 1, and 10 µM, and the cells were subsequently incubated for an additional 48 h. Following trypsinization for cell detachment, the cells were harvested through centrifugation, washed with 1 mL of PBS, stained in accordance with the manufacturer’s instructions, and subjected to analysis using a flow cytometer (FACSCalibur and BD CellQuest Pro™ software version 5.0, BD Bioscience, San Jose, CA, USA).

### 2.8. Cell Cycle Arrest

The processed cell specimens underwent a cold PBS wash and were subsequently preserved in 70% ethanol at −20 °C overnight. Following another round of PBS washing, the specimens were reconstituted in 0.5 mL of FxCycle™ PI/RNase Staining Solution (Invitrogen), which consisted of 50 μg/mL PI and 100 μg/mL RNase A. The mixture was then subjected to a 30 min incubation at room temperature in the absence of light. Subsequent analysis of the specimens was conducted utilizing a FACSCalibur flow cytometer (BD Bioscience, San Jose, CA, USA).

### 2.9. Cell-Derived Mouse Tumor Xenografts Model

BALB/c nude female mice (7 weeks old) were provided by ORIENT BIO Inc. (Seongnam, Republic of Korea). Mice were housed in a pathogen-free room at a controlled temperature (25 ± 2 °C) and humidity (65 ± 5%) and alternating 12 h light/dark cycles. After acclimation for 1 week, 100 µL of Matrigel containing 5 × 10^7^ SKOV3 cells was subcutaneously injected into the right flanks of BALB/c nude mice. We excluded the mice with a tumor size less than 150 mm^3^ four weeks after implantation. Mice with successful tumor formation were randomly divided into three groups (n = 10 for each group) as follows:

Part 1. OPT-0139 single treatment

Group 1: tumor control (implanted SKOV3 cells);Group 2: tumor cell implantation and intravenous low-dose OPT-0139 (5 mg/kg);Group 3: tumor cell implantation and intravenous high-dose OPT-0139 (20 mg/kg).

Part 2. OPT-0139 combination therapy with cisplatin

Group 1: tumor control (implanted SKOV3 cells)Group 2: tumor cell implantation and intravenous cisplatin (2 mg/kg);Group 3: tumor cell implantation and intravenous OPT-0139 (5 mg/kg);Group 4: tumor cell implantation and intravenous OPT-0139 and cisplatin (OPT-0139 5 mg/kg, cisplatin 2 mg/kg).

Tumor areas were measured once every 4 days and calculated as ([width (mm) × width (mm)] × length [mm])/2 using a Vernier caliper. OPT-0139 was dissolved in PBS and intravenously administered once every 3 days. For the continuous 4-week treatment, body weight was measured twice weekly. Mice were then euthanized, and the tumors were isolated. All animal experiments followed the guidelines of the Seoul National University Bundang Hospital Institutional Animal Care and Use Committee (IACUC). Institutional Review Board statement: The IACUC of the Seoul National University Bundang Hospital approved this study (No. BA-2111-332-010-02) (approval date: 26 November 2021).

### 2.10. Immunohistochemical Staining

Tumor tissues obtained from cell-derived xenograft mice underwent deparaffinization, rehydration, and dual washes with buffer. Subsequently, the slides were subjected to a 10 min incubation in a hydrogen peroxide block, followed by four washes with buffer to minimize nonspecific background staining associated with endogenous peroxidases. Primary antibodies were introduced and incubated in accordance with the manufacturer’s guidelines, followed by four additional buffer washes. The slides underwent treatment with a primary antibody enhancer, a 20 min incubation at room temperature, and four subsequent buffer washes. Following this, HRP Polymer was applied to the slides, leading to a 30 min incubation at room temperature and four more buffer washes. Lastly, the slides were incubated with hematoxylin as a chromogen and underwent four washes with deionized water.

### 2.11. Quantitative Real-Time PCR Analysis

Total RNA was isolated from the tumor samples using the TRIzol reagent (Invitrogen) as per the manufacturer’s guidelines. DNA-free RNA, in equal amounts, underwent reverse transcription to generate cDNA through the application of GoScript™ Reverse Transcriptase (Promega). Real-time PCR was conducted in a 25 μL reaction volume, utilizing 3 μL of a 1:10 cDNA dilution containing SYBR Green master mix (BioRad, Hercules, CA, USA) and specific primers for PCR targeting BRD4, cyclin-dependent kinase inhibitor 1A (p21), cyclin-dependent kinase inhibitor 1B (p27), MYC proto-oncogene, bHLH transcription factor (c-Myc), hypoxia-inducible factor 1 subunit alpha, vascular endothelial growth factor (VEGF), octamer-binding transcription factor 4 (OCT-4), Nanog homeobox (Nanog), b-cell lymphoma 2 (BCL-2), and Bcl-2-associated X (BAX) (Appendix A). All PCR reactions were performed using a Qiagen Rotor-Gene Q real-time PCR system, and fluorescence threshold values (Ct) were computed. Relative mRNA levels were determined by normalizing to 18s rRNA, and the outcomes were presented as fold differences in gene expression.

### 2.12. Immunoblot Analysis

Cell monolayers underwent a single wash with ice-cold saline before being lysed by the addition of PROPREP (50 mM Tris-HCl [pH 8.0], 150 mM NaCl, 1% Triton X-100, 0.5% sodium deoxycholate, and 0.1% SDS) along with protease inhibitors (iNtRON Biotechnology, Seongnam, Republic of Korea). The lysate was subjected to a 1 h incubation at 4 °C with gentle agitation, followed by harvesting using a cell scraper and transfer to a 1.5 mL tube. Subsequent centrifugation at 12,000× *g* for 10 min at 4 °C separated the lysate into supernatants, which were then collected and stored at −80 °C. Protein concentrations were determined through the bicinchoninic acid protein assay (Pierce, Thermo Fisher Scientific). The separation of total cell extracts occurred via 12% SDS-PAGE, followed by transfer onto PVDF membranes. Blocking of the membranes with 5% skimmed milk preceded their incubation with primary antibodies, appropriately diluted in a blocking solution. Signal visualization utilized the chemiluminescent substrate method and the Super Signal West Pico kit (Pierce, Thermo Fisher Scientific). To normalize loading, β-actin served as an internal control.

### 2.13. Statistical Analysis

Every experiment was conducted in triplicate and iterated at least three times. The statistical results are presented as the mean ± standard deviation (SD). To assess statistical variances, the Mann–Whitney U test was executed. All statistical assessments were two-sided, with a significance level set at *p* < 0.05 for all instances. The significance levels are denoted as follows: *, *p* < 0.05; **, *p* < 0.01; and ***, *p* < 0.001. Statistical analyses were performed using GraphPad Prism (GraphPad Software, version 9.4.1).

## 3. Results

### 3.1. BRD4 Overexpression and IC50 Value of OPT-0139 in Human Ovarian Cancer Cells

To evaluate BRD4 expression in ovarian cancer cell lines, we used the SKOV3, A2780, OVCAR3, and CAOV3 cell lines. HOSEpic was used as a control group. Western blot analysis was performed to compare BRD4 expression in each cell line, and relative mRNA levels were measured using quantitative real-time PCR. BRD4 expression was higher in SKOV3 and OVCAR3 cells than that in the other cell lines (Figure 1A,B). The subsequent experiments were conducted with high-grade serous ovarian adenocarcinoma cell lines SKOV3 and OVCAR3 because of their high BRD4 expression levels.

The half-maximal inhibitory concentration (IC50) of OPT-0139 was evaluated to determine its potency to reduce ovarian cancer cell line viability. SKOV3 and OVCAR3 cells were treated with 1 nM–100 μM of OPT-0139 for 48 h (Figure 1C,D). Cell viability was measured after 48 h. The IC50 value for the drug was 1.568 and 1.823 μM in the SKOV3 and OVCAR3 cell lines, respectively.

### 3.2. OPT-0139′s Inhibitory Effect on Cell Survival and Proliferation

To explore the inhibitory effect of OPT-0139 on cell viability and proliferation, Presto Blue and CCK-8 assays were performed. Based on the IC50 value of OPT-0139, the effective concentrations of the drug were 0, 0.01, 0.1, 0.5, 1, 5, and 10 μM. SKOV3 and OVCAR3 cells were treated with OPT-0139 at each concentration for 48 h to determine cell viability and proliferation (Figure 2A,B). In terms of cell viability assessment, the PrestoBlue reagent, utilizing the MTT assay, was employed. Additionally, the evaluation of cell proliferation was conducted using the CCK-8 assay, demonstrating a dose-dependent response with OPT-0139 in the context of investigating the anticancer effects of the BRD4 inhibitor. Consequently, cell viability decreased from 0.1 μM of OPT-0139 in SKOV3 and OVCAR3 cells (Figure 2A). Moreover, cell proliferation decreased dose-dependently from 0.5 μM (Figure 2B). OPT-0139 caused a dose-dependent decrease in ovarian cancer cell viability and proliferation.

In addition, immunocytochemistry was used to identify cell structure change and BRD4 expression. SKOV3 cells were treated with OPT-0139, and green fluorescence indicated BRD4, whereas red indicated mitochondrial staining to confirm the cell shape (Figure 2C). Treatment with OPT-0139 decreased BRD4 expression in the ovarian cancer cell lines (Figure 2C).

### 3.3. OPT-0139′s Induction of Apoptotic Cell Death

To identify the correlation between BRD4 inhibitor and apoptotic cell death induction, annexin V and PI staining was performed on the SKOV3 and OVCAR3 cell lines with OPT-0139 incubation for 48 h. Flow cytometry analysis of annexin V-PI staining showed that the apoptotic cell count, especially early apoptosis, increased dose-dependently in SKOV3 and OVCAR3 upon OPT-0139 treatment. The quantitative analysis illustrates that SKOV3 and OVCAR3 significantly increased apoptosis depending on OPT-0139 concentration. (*** *p* < 0.001) (Figure 3A).

A caspase-3 activity assay showed that OPT-0139 significantly induced caspase-3 activity in a concentration-dependent manner (Figure 3B). Western blot analysis was performed to evaluate apoptotic signaling protein expression. Western blot analysis showed that OPT-0139 decreased the anti-apoptotic molecule—Bcl-2. In contrast, the pro-apoptotic molecules’ expression of cleaved caspase-3 and PARP increased (Figure 3C). Thus, these results demonstrated that the BRD4 inhibitor activates the apoptosis signaling pathway and induces apoptosis.

### 3.4. OPT-0139′s Induction of Cell Cycle Arrest

To explore whether OPT-0139 causes cell cycle arrest in ovarian cancer cells, SKOV3 and OVCAR3 cells were treated with OPT-0139 and cell cycle analyses were performed by flow cytometry. OPT-0139 induced G1 phase arrest in both ovarian cancer cell lines. It decreased the G1 phase cell count and increased the G2/M phase cell count (Figure 4A,B). RT-PCR was performed to understand the cell cycle arrest pathway. In both ovarian cancer cell lines, SKOV3 and OVCAR3, BRD4 expression decreased, whereas CDKN1A (p21) gene expression and degradation of the cell cycle inhibitor CDKN1B/p27KIP1 (p27) increased in a dose-dependent manner. The expression of c-Myc also decreased (Figure 4C).

### 3.5. OPT-0139′s Inhibition of Tumorigenesis in a Mouse Xenograft Model

A xenograft mouse model was established to determine the potential therapeutic effect of OPT-0139 on tumor growth, and tumor volumes were measured every 3 days. We examined the effects of OPT-0139, a single treatment, on the tumor growth of SKOV3 cells in vivo; the SKOV3 cells are highly tumorigenic in nude mice. The female BALB/c nude mice (8 weeks old) were used as an animal model to prepare the mouse xenograft model. Each mouse was inoculated subcutaneously with 100 μL of Matrigel and human ovarian cancer SKOV3 cells (5 × 10^7^). We excluded the mice with a tumor size < 150 mm^3^. Mice were divided into three groups (n = 10) 30 days after inoculation and matched for tumor volume. Mice groups were treated with intravenous injection of OPT-0139 and were sacrificed after 4 weeks of treatment, and tumor tissues were collected. We monitored the tumor growth rate and body weight for each mouse group, and the OPT-0139 single treatments notably reduced the tumor growth rate in mice compared to that in the control groups (Figure 5A). BRD4 expressions were detected in mouse xenograft tumors using immunohistochemistry (Figure 5C). Furthermore, the single treatment of OPT-0139 significantly suppressed the final tumor weight and volume in a dose-dependent manner (Figure 5D). However, no impact on body weight was found in either of the mouse groups (Figure 5B,D). No mortality occurred among the subjects throughout the course of the experiment, and mice devoid of tumor growth were subsequently excluded from this study. These in vivo data strongly suggest that OPT-0139 single treatment significantly suppressed tumorigenesis in a mouse tumor xenograft model.

### 3.6. OPT-0139 Alters Hypoxia Signaling, Angiogenesis, Cancer Stemness, and Apoptosis Gene Expression Levels in Tumors

BRD4 controls the regulation of HIF-1a, a gene associated with hypoxia responses. As HIF-1a functions as a transcription factor overseeing the expression of various genes related to angiogenesis, glycolysis, and other adaptive mechanisms triggered by insufficient oxygen levels [20,21].

To assess whether OPT-0139 affects mRNA expression of hypoxia signaling, angiogenesis, cancer stemness, and apoptosis markers, total RNA was isolated from control, OPT-0139 5 or 20 mg/kg-injected mouse tumor and analyzed using semi-quantitative real-time-PCR. The mRNA expression levels of BRD4 and hypoxia signaling genes were markedly decreased (BRD4, ~50 and ~70%, Hif-1α, ~40 and ~70, respectively) in the OPT-0139 5 or 20 mg/kg-injected groups (Figure 6A,B). Furthermore, the mRNA expression levels of VEGF (angiogenesis), Oct-4, and Nanog (cancer stemness) were suppressed by ~50% for the OPT-0139 5 or 20 mg/kg-injected group (Figure 6C–E). Additionally, mRNA expression levels of Bcl-2 (anti-apoptosis) were decreased by 50 and 70% (Figure 6F), and the mRNA expression levels of BAX (anti-Bcl-2) were increased by ~2.5 fold for the OPT-0139 5 or 20 mg/kg-injected groups, respectively (Figure 6G). A significant difference was observed in every feature when comparing the differences between the control, low-dose, and high-dose groups. Even though the *p*-value was less than 0.5 between the low-dose and high-dose groups, they showed a statistically significant difference (Figure 6A–G). These results indicate the inhibition or activation effects of OPT-0139 on BRD4, hypoxia signaling, and tumor progression.

### 3.7. Combining OPT-0139 with Cisplatin in Ovarian Cancer Cells

To investigate the possible synergistic impact of OPT-0139 with cisplatin in ovarian cancer cells, the SKO3 and OVCAR3 cell lines were treated with 0.1 μM OPT-0139 with or without 10 μM cisplatin. Compared to using either agent, the combination of OPT-0139 with cisplatin showed that cell viability was reduced by about half in both cell lines (Figure 7A). Caspase-3 activity was performed with the same setting of 0.1 μM OPT-0139 with or without 10 μM cisplatin in two cell lines. The result showed that the combined use of OPT-0139 and cisplatin increased the activity of caspase-3 by 4–5 fold compared to the groups with independent use of each agent (Figure 7B). Cell viability and caspase-3 activity showed statistical differences between the four groups—control vs. OPT-0139-only vs. cisplatin-only vs. combination group (Figure 7A,B). Western blot analysis was performed with OPT-0139 and cisplatin combination setting to compare the effects of monotherapy of each drug and combination therapy. On a protein level, no significant differences could be observed when cell lines were treated with OPT-0139 and cisplatin alone. However, when both drugs were combined, apoptosis-related molecules’ expression showed the same trend but with a more pronounced effect. Expression of the anti-apoptotic molecule—Bcl-2—decreased, and expression of pro-apoptotic molecules—BAX, cleaved caspase-3, and PARP—increased (Figure 7C).

### 3.8. Combining OPT-0139 with Cisplatin in a Mouse Xenograft Model

Based on the results of Figure 5, we validated the additive effect of combining OPT-0139 with cisplatin in a mouse xenograft model using SKOV3 cells. Four experimental settings were created: DMSO as a control group, OPT-0139 as a BRD4 inhibitor group, cisplatin as a chemotherapy group, and OPT-0139 as a combination group. Compared to cisplatin alone, the tumor growth rate was significantly reduced when OPT-0139 was given in combination with cisplatin (Figure 8A). Tumor weight was measured in isolation from mice. Compared to the cisplatin-only group, the combination of OPT-0139 reduced tumor weight by approximately 90% and tumor volume by approximately 80% (Figure 8C,D). Statistical differences between the monotherapy and combination groups of OPT-0139 and cisplatin are shown (Figure 8C–E). In addition, the body weights of all mice did not decrease when measured to check for side effects (Figure 8B,E).

To determine if OPT-0139–cisplatin combination treatment alters the expression level of BRD4, hypoxia signaling, angiogenesis, cancer stemness, and apoptosis marker genes, total RNA was isolated from the control, OPT-0139, cisplatin, OPT-0139/cisplatin-injected mouse tumor, and then analyzed by semi-quantitative real-time PCR. The expression level of BRD4 and hypoxia signaling genes were significantly decreased (BRD4, ~35 and ~20%; Hif-1α, ~55 and ~60%), respectively, in the OPT-0139- or OPT-0139/cisplatin-injected groups (Figure 9A,B), but no significant changes were observed in expression levels in the cisplatin-injected group. Expression levels of VEGF (~50% and ~80%), Nanog (~40% and ~60%), and Oct-4 (~40% and ~45%) were suppressed in the OPT-0139 or OPT-0139/cisplatin-injected group (Figure 9C–E). Furthermore, mRNA expression levels of Bcl-2 were reduced by ~55 and ~70% (Figure 9F), and the mRNA expression levels of BAX were induced by ~2.5 and ~2.0 fold in the OPT-0139 or OPT-0139/cisplatin-injected groups, respectively (Figure 9G). Every feature showed statistically significant differences (*p*-value < 0.001) (Figure 9A–G). Our results revealed that OPT-0139 has the potential to be developed as an effective therapy to overcome ovarian cancer.

## 4. Discussion

In the present study, we demonstrated an anticancer effect of the novel BRD4 inhibitor OPT-0139 in combination with the conventional cytotoxic agent cisplatin in vitro and in vivo. Laboratory findings were conducted with two ovarian cancer cell lines, SKOV3 and OVCAR3, and mouse xenograft models. We observed inhibition of cell survival and proliferation and induction of cell cycle arrest and apoptosis in the OPT-0139 and cisplatin groups. Our key findings suggest that OPT-0139 induces several anticancer effects, such as cell viability decrease and induction of apoptosis, independently and in combination with cisplatin.

Studies have revealed that BET proteins regulate cell cycles and promote inflammatory cytokine production [10,21]. Deregulation of BET proteins, particularly BRD4, has been implicated in the development of various diseases, especially cancer. BRD4 inhibitors have demonstrated promising results in both preclinical and clinical studies against various cancer types, including leukemia, lymphoma, and solid tumors such as breast, lung, and prostate cancers [22,23].

HIF-1a is one of the genes regulated by BRD4. Since HIF-1a is a transcription factor that regulates the expression of numerous genes involved in hypoxia responses, inhibiting its stability or activity can lead to reduced expression of HIF-1a target genes. This includes genes involved in angiogenesis, glycolysis, and other adaptive responses to low oxygen levels [20,21]. HIF-1a and c-Myc expression are correlated with cell cycle arrest and cell proliferation [24]. In ovarian cancer, c-Myc is overexpressed [25].

BRD4 is unevenly distributed within cancer cells, with higher concentrations observed for some important oncogenes, such as c-Myc. It then promotes the activation of these genes, leading to increased cell growth [26,27,28]. Our study agent, OPT-0139, is designed to induce anticancer effects by interfering with HIF-1a signaling by inhibiting BRD4. Subsequently, HIF-1a downregulated important cell cycle genes, such as c-Myc. We confirmed the signaling pathway with HIF-1a and c-Myc using RT-PCR and real-time PCR (Figure 4C, Figure 6 and Figure 9).

Inducing cell proliferation by promoting the transition from G1 to S phase during cell cycle progression is one of the most characteristic functions of c-Myc, a feature associated with its pro-tumor activity. Several mechanisms account for this, including repression of the CDKN2B (p15) and CDKN1A (p21) genes and degradation of the cell cycle inhibitor CDKN1B/p27KIP1 (p27) [29]. p21 and p27 act as universal inhibitors of Cyclin-dependent kinases (CDKs) by inhibiting various CDK-cyclin complexes [30,31]. In ovarian cancer, the expression of these two proteins is associated with survival [32]. RT-PCR of the proteins involved was performed to determine if they induced cell cycle arrest, and it was found that p21 and p27 expressions increased in a concentration-dependent manner after OPT-0139 treatment, whereas c-Myc expression decreased (Figure 4C). By identifying these proteins, we were able to predict the pathway indirectly.

Few studies have attempted to adapt BRD4 inhibitors to treat ovarian cancer cells [17,18,33,34,35], and most of them have focused on the relationship between PARP inhibitors and BET inhibitors or the correlation between BRD4 expression and basic function in ovarian cancer [18,33,34]. The heterogeneity of ovarian cancer is one of the reasons why, despite several studies, there has not been a game-changer in its treatment. Ovarian cancer carcinogenesis is heterogeneous and differs from other cancers. Its complexity makes treatment difficult, and the recurrence rate is high [17,19]. Therefore, novel treatment options are needed, and epigenetic approaches have emerged. Among epigenetic anticancer drugs, BRD4 inhibitors have received attention owing to their various functions, such as decreased cell viability, cell proliferation, and apoptosis induction [9,36]. While there is a lack of extensive data regarding the impact of BRD4 inhibitors on ovarian cancer in the existing literature, our research aligns with the outcomes of previous studies [17].

Our data are clinically relevant as we have revealed the possibility of overcoming the limitations of ovarian cancer treatment and the additive effect of a BRD4 inhibitor with the standard anticancer cisplatin. However, this study had some limitations in using mouse heterotopic xenograft models.

In Figure 7, the results showed a relatively poor response to cisplatin in the SKOV3 cell line (Figure 7A,C); this result was unexpected since we did not use a platinum-resistant SKOV3 cell line. The recent doubts raised about the histological derivation of SKOV3 cells pose difficulties in extending the conclusions of this study [37]. It is unclear whether this result is a characteristic of the SKOV3 cell line or the heterotopic xenograft mouse model used in this experiment or due to the concentration of cisplatin used. In addition, there have been multiple suggestions about the IC50 of cisplatin in the SKOV3 cell line. Thus, it is advisable to interpret the results of the experiments performed on the SKOV3 cell line cautiously and plan any subsequent experiments with these results in mind [38,39].

The subcutaneous model can easily evaluate subcutaneous tumor masses and is easy for tumor growth monitoring [40]. However, most ovarian cancer cells in patients grow in the abdominal cavity and rarely metastasize to the subcutaneous tissues. Since intraperitoneal or orthotopic models more reflect the tumor microenvironment than subcutaneous models, we can cautiously infer that cisplatin chemotherapy might not be as effective as expected in the subcutaneous SKOV3 tumor models [40].

It is evident that there remains a substantial scope for further understanding the implications of BRD4 in the context of cancer. Our current study, while informative, is not without limitations. Notably, we did not conduct a tolerability test for both cisplatin and OPT-0139, and the sample numbers in our in vivo study with xenograft mice were not uniform. Acknowledging these inherent limitations, our research endeavors are dedicated to investigating the efficacy of BRD4 inhibitors in the context of ovarian cancer. This decision is underpinned by the observed anticancer effects associated with BRD4 inhibitors across multiple cancer types, demonstrating effectiveness both independently and in conjunction with conventional agents.

In this study, we confirmed the tendency of the drug (OPT-0139) to exhibit additive effects with cisplatin. Compared to OPT-0139 used alone, the combination of cisplatin and OPT-0139 displayed a greater decrease in BRD4 expression and tumor cell viability; cell viability decreased by about 40% with the use of cisplatin alone and by almost 60% with the combined use of OPT-0139 and cisplatin (Figure 7A). Changes in the expression of apoptosis signaling molecules, including decreased cell viability and caspase-3 activity, suggest that the combination therapy has an additive effect.

## 5. Conclusions

Our study has validated a new BRD4 inhibitor, OPT-0139, as a promising anticancer agent for ovarian cancer treatment. Furthermore, once validated in preclinical trials, initial assessments showed that treatment effectiveness could be improved when combined with conventional cytotoxic agents, offering hope to poor responders to current treatments. Further research into the role of BRD4 inhibitors in inhibiting metastasis in ovarian cancer, as shown in breast cancer, would help treat ovarian cancer [9].

## Figures and Tables

**Figure 1 cancers-16-00959-f001:**
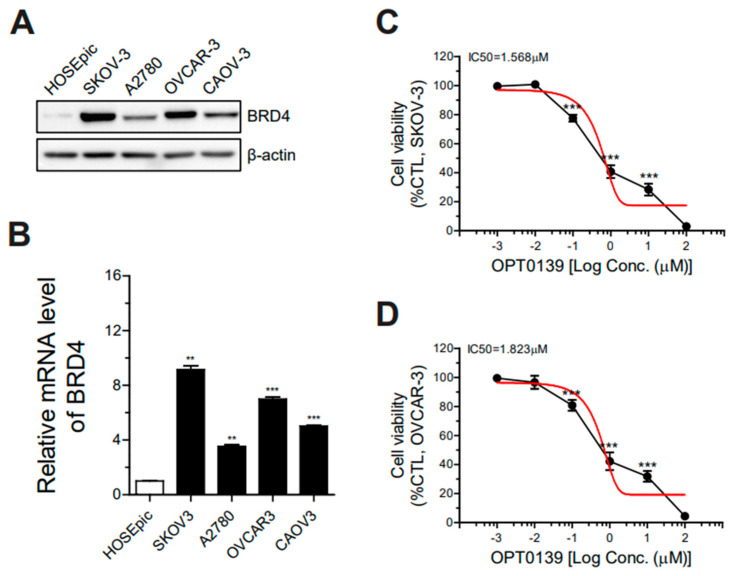
BRD4 expression in various ovarian cancer cell lines, SKOV3, A2780, OVCAR3, CAOV3, and HOSEpic. (**A**) Western blots (n = 1) and (**B**) relative mRNA levels of each cell line were examined (n = 4). (**C**,**D**) IC50 value of OPT-0139 in SKOV3, and OVCAR3 cells (n = 8, the cell experiment was repeated twice, each time using four independent cell cultures) (** *p* < 0.01, *** *p* < 0.001 compared to the control group).

**Figure 2 cancers-16-00959-f002:**
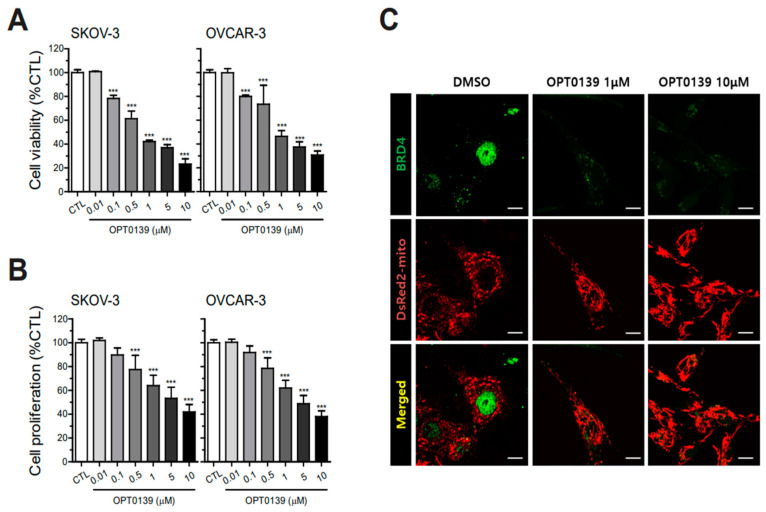
Inhibition of cancer cell survival and proliferation by OPT-0139 (OPT-0139 treatment concentration between 0.01 and 10 μM) (**A**) cell viability (n = 8, the cell experiment was repeated twice, each time using four independent cell cultures) and (**B**) cell proliferation (n = 8, the cell experiment was repeated twice, each time using four independent cell cultures) (*** *p* < 0.001 compared to the control group). (**C**) Immunocytochemistry in SKOV3. Green fluorescence: BRD4, red fluorescence: mitochondria (n = 3) (scale bar: 1 μm).

**Figure 3 cancers-16-00959-f003:**
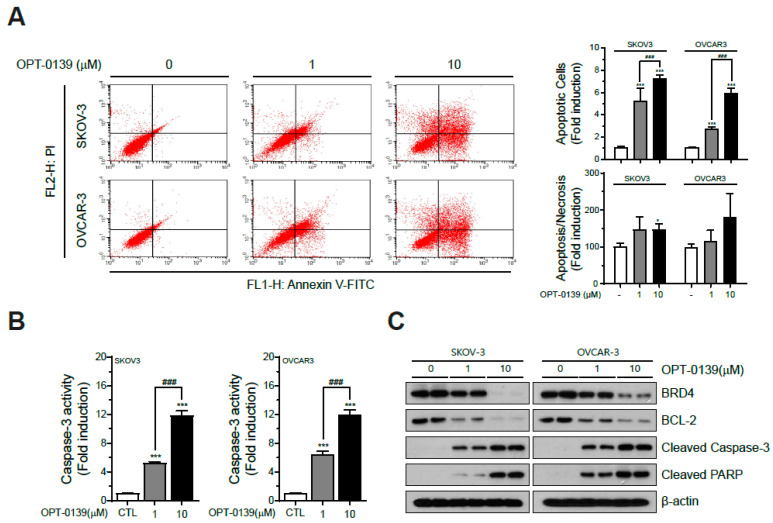
Induction of apoptotic cell death by OPT-0139 (**A**) apoptosis (n = 3) and (**B**) caspase-3 activity analysis of SKOV3 and OVCAR3 cells were conducted following OPT-0139 treatment (n = 8, the cell experiment was repeated twice, each time using four independent cell cultures) (* *p* < 0.05, *** *p* < 0.001 compared to the control group, ### *p* < 0.001 compared to the experimental groups); (**C**) Western blot (n = 3).

**Figure 4 cancers-16-00959-f004:**
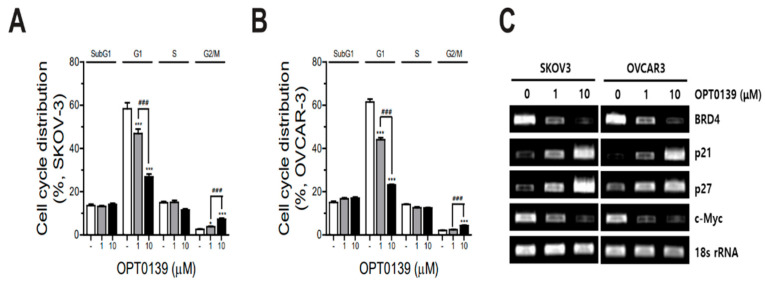
Cell cycle arrest induction by OPT-0139: OPT-0139 affects cell cycle phase distribution of (**A**) SKOV3 (n = 3) and (**B**) OVCAR3 cells (n = 3) (* *p* < 0.05, *** *p* < 0.001 compared to the control group, ### *p* < 0.001 compared to the experimental groups); (**C**) RT-PCR analysis for understanding the cell cycle arrest pathway (n = 4, with the sample number doubled by sampling two per tumor). Abbreviations: p21, CDKN1A; p27, CDKN1B/p27KIP1; c-Myc, MYC proto-oncogene; bHLH, transcription factor.

**Figure 5 cancers-16-00959-f005:**
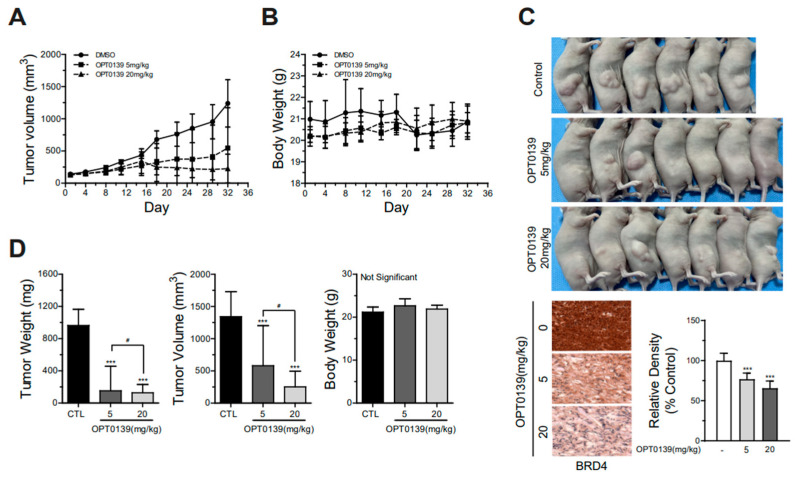
Anticancer effect of OPT-0139 in mouse xenograft models with SKOV3 cell line injection: (**A**) Calculated tumor volume (DMSO group n = 6, OPT-0139 5 mg group n = 7, and OPT-0139 20 mg group n = 7). (**B**) Body weight. (**C**) Gross tumors and BRD4 immunohistochemical staining analysis in SKOV3 cell tumors for each representative group of mice (n = 4). Please note that mice with no tumor growth were excluded from the analysis, accounting for the variation in sample numbers among the three groups. (**D**) Tumor weight and tumor volume of isolated tumors from mice and Body weight of mice (DMSO group n = 6, OPT-0139 5 mg group n = 6, and OPT-0139 20 mg group n = 5) (*** *p* < 0.001 compared to the control group, # *p* < 0.05 compared to the experimental groups).

**Figure 6 cancers-16-00959-f006:**
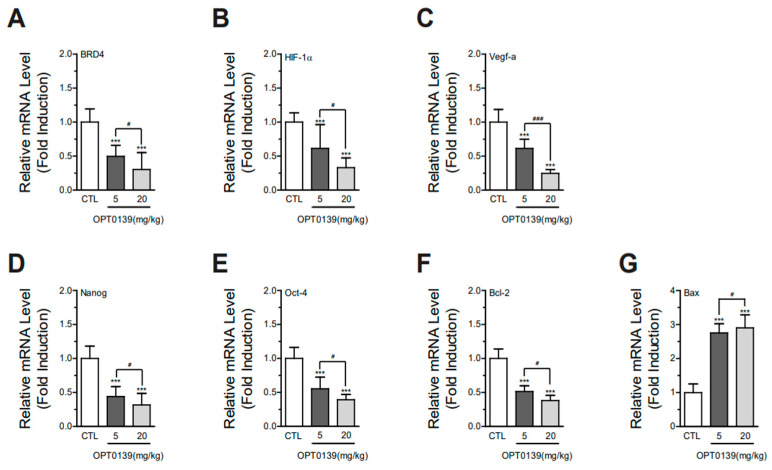
Relative mRNA level measurement with OPT0139 treatment (CTL vs. 5 mg/kg vs. 20 mg/kg): (**A**) BRD4, bromodomain-containing protein; (**B**) HIF-1a, hypoxia-inducible factor; (**C**) Bcl-2, b-cell lymphoma 2; (**D**) BAX, Bcl-2-associated X; (**E**) VEGF-a, vascular endothelial growth factor; (**F**) Nanog, Nanog homeobox; (**G**) Oct-4, octamer-binding transcription factor 4 (DMSO group n = 12, OPT-0139 5 mg group n = 12, and OPT-0139 20 mg group n = 10, with the sample number doubled by sampling two per tumor) (*** *p* < 0.001 compared to the control group, # *p* < 0.05, ### *p* < 0.001 compared to the experimental groups).

**Figure 7 cancers-16-00959-f007:**
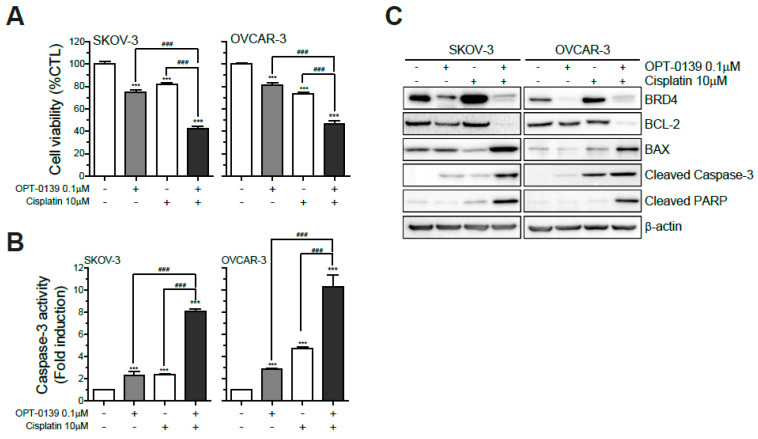
Additive effect of combining OPT-0139 with cisplatin in ovarian cancer cells: (**A**) cell viability (n = 8, the cell experiment was repeated twice, each time using four independent cell cultures), (**B**) caspase-3 activity (n = 8, the cell experiment was repeated twice, each time using four independent cell cultures), and (**C**) Western blot (n = 3). Abbreviations: BRD4, bromodomain-containing protein 4; Bcl-2, b-cell lymphoma 2; BAX, Bcl-2-associated X; cleaved PARP, cleaved poly ADP-ribose polymerase (*** *p* < 0.001 compared to the control group, ### *p* < 0.001 compared to the experimental groups).

**Figure 8 cancers-16-00959-f008:**
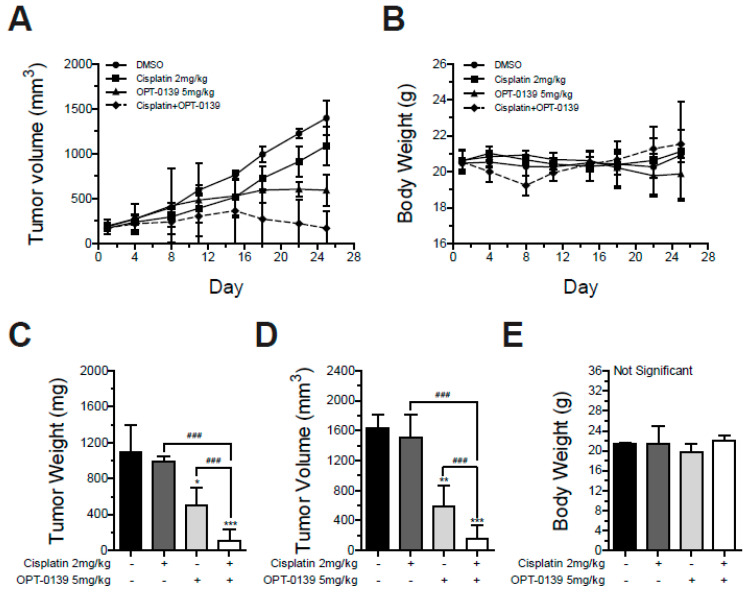
Additive effect of combining OPT-0139 with cisplatin. (**A**,**B**) Calculated tumor volume and Body weight change (DMSO group n = 3, cisplatin-only group n = 3, OPT-0139 5 mg group n = 5, and cisplatin + OPT-0139 group n = 5). (**C**,**D**) Tumor weight and tumor volume of isolated tumors from mice. (**E**) Body weight change (DMSO group n = 3, cisplatin-only group n = 3, OPT-0139 5 mg group n = 5, and cisplatin + OPT-0139 group n = 3) (* *p* < 0.05, ** *p* < 0.01, *** *p* < 0.001 compared to the control group, ### *p* < 0.001 compared to the experimental groups; N.S., not significant).

**Figure 9 cancers-16-00959-f009:**
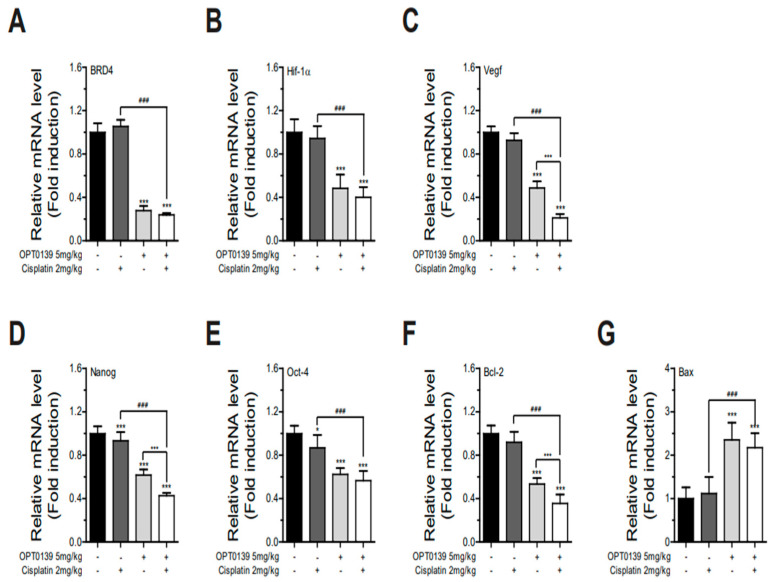
Relative mRNA levels between the control, cisplatin-only, OPT-0139-only, and OPT-0139 and cisplatin combination groups: (**A**) BRD4; (**B**) HIF-1a; (**C**) Bcl-2; (**D**) BAX; (**E**) VEGF-a; (**F**) Nanog; (**G**) Oct-4 (CTL group n = 6, cisplatin-only group n = 6, OPT-0139 5 mg group n = 10, and cisplatin + OPT-0139 group n = 6, with the sample number doubled by sampling two per tumor) (* *p* < 0.05, *** *p* < 0.001 compared to the control group, ### *p* < 0.001 compared to the experimental groups).

## Data Availability

Data are contained within this article or in the Appendix A.

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
