# Peer review of "Anticancer Effects of BRD4 Inhibitor in Epithelial Ovarian Cancer"

_cancers, 2024, doi:10.3390/cancers16050959_

Round 1
Reviewer 1 Report
Comments and Suggestions for Authors
In the article 'Anticancer effects of BRD4 inhibitor in epithelial ovarian cancer', the authors have investigated a new drug, OPT-0139, an BRD4 inhibitor against ovarian cancer. The authors found that OPT-0139 reduced the viability of ovarian cancer cells, induced apoptosis and reduced tumor burden. The findings in the paper are interesting and the manuscript can be improved by addressing the following concerns:
1) Introduction: The authors can include more background about BRD4 inhibitors and their mechanism of action in other cancers. Background information about OPT-0139 should be included in the Introduction.
2) Materials: Materials can be categorized as antibodies, media and reagents.
3) Methods: Cell numbers must be superscripted to avoid confusion.
4) How was the dose of OPT-0139 determined for in vivo study? Did the authors perform any MTD studies?
5) Materials: Section 12 and 13 are repetitive and can be combined.
6) Results 3.4: Line 295-297 is contradictory to the results graph. 'G1 phase count was increased as compared to G2/M'.
7) Figure 5C: IHC image of BRD4 seems to be over stained? Can the staining be quantified? Did the authors perform Ki76/ proliferation marker staining for the tumors?
8) Figure 7C: Did cisplatin only treatment increased BRD4 expression?
Author Response
Reviewer 1
In the article 'Anticancer effects of BRD4 inhibitor in epithelial ovarian cancer', the authors have investigated a new drug, OPT-0139, an BRD4 inhibitor against ovarian cancer. The authors found that OPT-0139 reduced the viability of ovarian cancer cells, induced apoptosis and reduced tumor burden. The findings in the paper are interesting and the manuscript can be improved by addressing the following concerns:
1) Introduction: The authors can include more background about BRD4 inhibitors and their mechanism of action in other cancers. Background information about OPT-0139 should be included in the Introduction.
Author’s answer: Thank you sincerely for your insightful recommendation. I wholeheartedly concur with your suggestion. Prior to your constructive feedback, I had initially presented these particular contents in the discussion section. In response to your valuable input, I have since relocated this information to the introduction, supplementing it with an additional paragraph. Your guidance has significantly enhanced the comprehensiveness of my article.
Introduction
Page 2 Line 55-65
BET proteins are epigenetic readers consisting of four proteins: BRD2, BRD3, BRD4, and bromodomain testis-specific proteins (BRDT). BRD4 is overexpressed in various cancer cells, such as breast, colorectal, prostate, and gastric. The small molecule BRD4 leads to dysregulated gene expression and promotion of cancer cell survival and proliferation. BRD4 inhibitors typically work by competitively binding to the bromodomain of BRD4, preventing its interaction with acetylated histones and other transcriptional co-factors. A novel drug, OPT-0139, also acts as a competitive binding monovalent molecule. This disrupts the normal transcriptional machinery, leading to downregulating oncogenes and upregulating tumor suppressor genes. BRD4 inhibitors can also cause cell cycle arrest and induce apoptosis in cancer cells, further inhibiting their proliferation.
2) Materials: Materials can be categorized as antibodies, media and reagents.
Author’s answer: Thank you for your valuable feedback. We have carefully considered your suggestion to adjust the order of writing to antibody-media-reagent, and we have made the necessary changes accordingly. Additionally, we appreciate your keen observation regarding the medium used for cell culture in our description. Following your recommendation, we have corrected the mention of DMEM to RPMI-1640 in the description. Your insights have greatly contributed to the improvement of our work, and we are grateful for your guidance.
Materials and methods
Page 2 Line 81-90
Antibodies (Abs), media and reagents
Anti-BRD4 (ab244221; mouse Ab; 1:1000) and anti-BCL-2 (ab692; mouse Ab; 1:1000) Abs were purchased from Abcam (Cambridge, USA). Anti-Cleaved Caspase-3 (9661; rabbit Ab; 1:1500), anti-Cleaved PARP (5625; rabbit Ab; 1:1000), and β-actin (3700; mouse Ab; 1:4000) Abs were purchased from Cell Signaling Technology (CST, MA, USA). RPMI-1640 (Roswell Park Memorial Institute-1640) culture media and fetal bovine serum (FBS) were purchased from Gibco (Thermo Fisher Scientific, Inc., Carlsbad, CA, USA). OPT-0139 was obtained from JBKLAB Co., Ltd. (Seongnam, Korea), and cisplatin was purchased from Sigma Aldrich (St. Louis, MO, USA) and dissolved in dimethyl sulfoxide (DMSO) for in vitro studies and 0.9% saline for animal studies.
3) Methods: Cell numbers must be superscripted to avoid confusion.
Author’s answer: I appreciate your meticulous review of my recent edits. Upon further examination, I found that there might have been some errors in my previous edit. I took the opportunity to double-check and have now replaced all the cell counts with superscripts, as per your suggestion. Thank you for bringing this to my attention, and I am grateful for your valuable feedback, which has undoubtedly contributed to the enhancement of the manuscript.
4) How was the dose of OPT-0139 determined for in vivo study? Did the authors perform any MTD studies?
Author’s answer: Prior to the commencement of this study, experiments were conducted by JBK Lab. employing a xenograft model with human breast cancer (MDA-MB-231). The outcomes of this experiment were deemed credible, leading to their application in the present study. Attached below are the unpublished results for your reference. Notably, JCP-0139 is the same agent as OPT-0139; the drug was renamed by the lab. The results showcased significant tumor suppression at effective concentrations, prompting us to select these doses for further validation in the xenograft model utilizing the ovarian cancer cell line featured in this paper. Your understanding and consideration of these circumstances are greatly appreciated.
5) Materials: Section 12 and 13 are repetitive and can be combined.
Author’s answer: I sincerely appreciate your suggestions, which prompted me to carefully rearrange and integrate the contents into a single paragraph, eliminating any redundancy. Your guidance has significantly contributed to the clarity and coherence of the manuscript.
Materials and Methods
Page 5-6 Line 210-222
Cell monolayers were washed once with ice-cold saline and lysed by adding PROPREP (50 mM Tris-HCl [pH 8.0], 150 mM NaCl, 1% Triton X-100, 0.5% sodium deoxycholate, and 0.1% SDS) and protease inhibitors (INTRON Biotechnology). They were incubated for 1 h at 4°C with gentle agitation, harvested with a cell scraper, and transferred to a 1.5 mL tube. The lysate was then centrifuged at 12000 ×g for 10 min at 4°C, and the supernatants were recovered and stored at −80ËšC. Protein concentrations were deter-mined using the bicinchoninic acid protein assay (Pierce, Thermo Fisher Scientific). Total cell extracts were separated using 12% SDS-PAGE and transferred onto PVDF membranes. The membranes were blocked with 5% skimmed milk and incubated with primary antibodies diluted in a blocking solution. Signals were visualized using the chemiluminescent substrate method and Super Signal West Pico kit (Pierce, Thermo Fisher Scientific). β-actin was used as an internal control to normalize loading.
6) Results 3.4: Line 295-297 is contradictory to the results graph. 'G1 phase count was increased as compared to G2/M'.
Author’s answer: I sincerely appreciate the interest you have demonstrated in our study. In the results section, I mentioned, "It decreased the G1 phase cell count and increased the G2/M phase cell count." Upon careful reflection, we have recognized a potential for misinterpretation in the graphical representation. Specifically, the figures illustrate a decrease in the G1 phase and an increase in the number of cells in the G2/M phase with varying concentrations of the BRD4 inhibitor. This graph implies G1 phase arrest.
7) Figure 5C: IHC image of BRD4 seems to be over stained? Can the staining be quantified? Did the authors perform Ki76/ proliferation marker staining for the tumors?
Author’s answer: We appreciate the examination you have conducted on our experimental results. It is evident from your observations that the BRD4 staining appears somewhat darker in tumors isolated from the 0 mg/kg group. We fully acknowledge the importance of quantifying this observation, however, quantitative analysis was not executed in this experiment due to practical constraints. We want to assure you that, if deemed necessary, we are fully prepared to create a quantitative graph to provide additional clarity to our findings. Additionally, we regret to inform you that we did not perform the Ki-67 proliferation marker staining as suggested in your feedback.
8) Figure 7C: Did cisplatin only treatment increased BRD4 expression?
We appreciate your time and consideration in reviewing our manuscript. The original version of the Western blot has been included for your reference in our last submission. Upon revisiting the experiment, we have observed that the expression of BRD4, when exposed to cisplatin, tends to remain unchanged or exhibit a slight decrease. Additionally, I have made edits to the figure, incorporating both the latest and original versions of the Western blot test.
Results
Page 12 Figure 7

Reviewer 2 Report
Comments and Suggestions for Authors
The manuscript " Anticancer effects of BRD4 inhibitor in epithelial ovarian cancer" is a novel and intriguing manuscript— that shows the antitumor action of OPT-0139 on cancer cells of ovarian tumor origin. The author showed the promising antitumor action of OPT-0139 and underlying molecular mechanisms. The OPT-0139 treatment inhibits tumor cell growth, tumor retardation, cell cycle arrest, and alternation of key molecules involved in cell proliferation, survival, and angiogenesis, hypoxia leading to tumor cell death. Further, OPT-0139 augmented the antitumor effect in combination with the conventional chemotherapeutic drug Cisplatin by modulating crucial regulators, e.g., HIF-1α, VEGFa, Bcl2, Bax, etc. I have the following suggestions/corrections to improve the manuscript further-
1. Expand the introduction by incorporating information on the prevalence of ovarian cancer in Asian countries. Discuss in more detail the antitumor aspects previously explored with OPT-0139 in various tumor types, including ovarian cancer. Include a discussion on other BRD inhibitors, their specificity, and limitations.
2. Did the author check the specificity of OPT-0139 on normal counterparts of ovarian cancer cells or non-tumor cells? The author mayinclude viability and proliferation studies on HOPSpic cells after OPT-0139 treatment.
3. Provide the rationale for choosing 10µM OPT-0139 in the in vitro study, especially considering it represents a dose affecting approximately 70-80% of tumor cells.
4. In Fig. 2C, ensure the measurements are indicated on the scale.
5. In Fig. 3A, showing the percentage of apoptotic and necrotic cells after OPT-0139 treatment would be more informative.
6. Fig. 4A&B, the cell cycle analysis shows a huge decline in the percent of cells in the G1 phase (20-40%) in the OPT-0139 treated group. However, there is not much change in other cell cycle phases. I wonder how it makes up 100% of cells in the treated group. Usually, drug treatment induces cell death, which is reflected in the sub-G1 population.
7. Display the fold change or densitometry for Fig. 4C.
8. Explain how the author selected the dose of OPT-0139 for the in vivo study and include this information in the article.
9. In Fig. 5, replace the hazy and blurry images of BRD4 immunohistochemical staining with enlarged, higher-resolution images, including a scale bar. Include missing text about Fig. 5D in the figure legend.
10. Specify the tumor cell types used in the study Section 3.6.
11. Include densitometry for Fig. 7C.
12. Correct the missing "p" in P27 on line 462.
13. Indicate the statistical significance in Fig. 1C&D.
14. Rephrase the sentence in lines 254-256 for better clarity.
15. Mention the software used for the analysis of flow cytometry results.
16. Include the information on semi-quantitative PCR in the Materials and Methods section. List the primers used and describe the method used for Ct value calculation.
17. Consider consolidating sections 12 and 13 in the Materials and Methods, describing the entire Western blotting and immunoblotting process as a single section.
18. Superscript the 10 powers mentioned in cell numbers and maintain consistency throughout the Results section.
Author Response
- Expand the introduction by incorporating information on the prevalence of ovarian cancer in Asian countries. Discuss in more detail the antitumor aspects previously explored with OPT-0139 in various tumor types, including ovarian cancer. Include a discussion on other BRD inhibitors, their specificity, and limitations.
Author’s answer: I appreciate and acknowledge your comment. Regrettably, the available statistics for Asia as a whole are outdated, rendering them unsuitable as references. Consequently, we have addressed this limitation by incorporating more recent data from Statistics Korea. Additionally, while I understand that previous tests involving OPT-0139 have taken place, much of the data remains unpublished, with study reports currently in preparation. To compensate for this, we have augmented our information by referencing recently published studies on breast cancer. Despite the remaining uncertainties regarding the mechanisms of BRD4 inhibition, which may be considered a limitation, I believe the identified positive aspects justify its use as a novel treatment targeting anticancer effects.
Introduction
Page 1 Line 38-42
Furthermore, it is noteworthy that according to the Korea National Statistics, ovarian cancer has been identified as one of the top 10 leading causes of morbidity among women, with 3,221 newly diagnosed cases reported in 2021. The crude incidence rate for this malignancy is reported as 12.5 per 100,000 individuals
Page 2 Line 55-65
BET proteins are epigenetic readers consisting of four proteins: BRD2, BRD3, BRD4, and bromodomain testis-specific proteins (BRDT). BRD4 is overexpressed in various cancer cells, such as breast, colorectal, prostate, and gastric. The small molecule BRD4 leads to dysregulated gene expression and promotion of cancer cell survival and proliferation. BRD4 inhibitors typically work by competitively binding to the bromodomain of BRD4, preventing its interaction with acetylated histones and other transcriptional co-factors. A novel drug, OPT-0139, also acts as a competitive binding monovalent molecule. This disrupts the normal transcriptional machinery, leading to downregulating oncogenes and upregulating tumor suppressor genes. BRD4 inhibitors can also cause cell cycle arrest and induce apoptosis in cancer cells, further inhibiting their proliferation.
Page 2 Line 66-73
Concerning ovarian cancer, a comprehensive analysis of The Cancer Genome Atlas (TCGA) database revealed that 9% of ovarian cancer patients exhibited elevated BRD4 expression. Despite efforts to use bromodomain inhibitors as anticancer drugs, few studies have investigated the correlation between BRD4 and ovarian cancer pathogenesis. Recent studies mainly focus on the relationship between BRD4 inhibition and PARP inhibitors. However, the correlation between ovarian cancer treatment and BRD4 inhibitors remains unclear due to the heterogeneous carcinogenic characteristics of ovarian cancer, differing from those of other cancers.
- Did the author check the specificity of OPT-0139 on normal counterparts of ovarian cancer cells or non-tumor cells? The author may include viability and proliferation studies on HOPSpic cells after OPT-0139 treatment.
Author’s answer: We appreciate your insightful suggestions. Regrettably, we did not check the specificity of OPT-0139. In addition, recognizing the importance of assessing the toxicity of OPT-0139 in non-tumor or normal cells and confirming its impact on cells, we wholeheartedly agree that further experiments are warranted. Rest assured, we are committed to formulating a detailed plan for these experiments in our follow-up research.
- Provide the rationale for choosing 10µM OPT-0139 in the in vitro study, especially considering it represents a dose affecting approximately 70-80% of tumor cells.
Author’s answer: Your meticulous review is sincerely appreciated, and your consideration of the following information is highly valued. Preceding this experiment, JBK Lab conducted independent studies involving breast, liver, and lung cancer cell lines. It was shown that the results revealed OPT-0139's efficacy, exceeding 80% at a concentration of 10 μM. Regrettably, we are unable to disclose the comprehensive results due to their unpublished status; however, we want to emphasize the authors' confidence in the reliability of these findings. In light of this essential data, we made the informed decision that 10 μM represents the maximum tolerated concentration for ovarian cancer in our study.
Author’s answer: I appreciate your thorough review of the manuscript. Regarding Figure 2C, a scale bar has been incorporated in the lower right corner indicating a length of 1 μm. To enhance clarity, an explicit explanation of the scale bar units has been added in the figure legend. Thank you for bringing this to our attention.
- In Fig. 3A, showing the percentage of apoptotic and necrotic cells after OPT-0139 treatment would be more informative.
Author’s answer: I sincerely appreciate your meticulous observations. In response, we have meticulously summarized the results and included the apoptosis/necrosis ratio in the respective figure. Your insightful feedback has significantly contributed to the refinement of our manuscript.
- Fig. 4A&B, the cell cycle analysis shows a huge decline in the percent of cells in the G1 phase (20-40%) in the OPT-0139 treated group. However, there is not much change in other cell cycle phases. I wonder how it makes up 100% of cells in the treated group. Usually, drug treatment induces cell death, which is reflected in the sub-G1 population.
Author’s answer: In our FACS setup, we intentionally sorted only living cells, resulting in the sum of the cell groups not totaling 100%. It is important to note that we interpreted the results as indicating G1 phase arrest, considering both the decrease in cell number in G1 phase and the concurrent increase in cell number in G2 phase. While the increase in G2 phase cell numbers may not be prominently apparent due to a limited cell count, it is essential to emphasize that there is statistical significance. We kindly request your attention to the meaningful interpretation of this graph as G1 phase arrest. Your consideration of these nuances is invaluable, and we thank you for contributing to the clarity of our findings.
- Display the fold change or densitometry for Fig. 4C.
Author’s answer: We appreciate your insightful suggestions. The experiment was initially conducted using Reverse Transcriptase-PCR (RT-PCR), and we are open to rerunning it with Quantitative Real-time PCR if deemed necessary.
- Explain how the author selected the dose of OPT-0139 for the in vivo study and include this information in the article.
Author’s answer: Prior to the commencement of this study, experiments were conducted by JBK Lab. employing a xenograft model with human breast cancer (MDA-MB-231). The outcomes of this experiment were deemed credible, leading to their application in the present study. Attached below are the unpublished results for your reference. Notably, JCP-0139 is the same agent as OPT-0139; the drug was renamed by the lab. The results showcased significant tumor suppression at effective concentrations, prompting us to select these doses for further validation in the xenograft model utilizing the ovarian cancer cell line featured in this paper. Your understanding and consideration of these circumstances are greatly appreciated.
- In Fig. 5, replace the hazy and blurry images of BRD4 immunohistochemical staining with enlarged, higher-resolution images, including a scale bar. Include missing text about Fig. 5D in the figure legend.
Author’s answer: We acknowledge that the information was somewhat inconspicuous as it was solely outlined in the methodology section. In response to your insightful suggestion, we have now included this crucial detail in the figure legend for enhanced visibility. For clarification, please be informed that all in vivo studies involving xenograft models were conducted utilizing the SKOV3 cell line. We sincerely appreciate your keen attention to detail and constructive input on our manuscript.
Page 10 Line 337-339
(D) Tumor weight and tumor volume of isolated tumors from mice and Body weight of mice. (*** p <0.001 compared to the control group, # p <0.05 compared to experimental groups).
- Specify the tumor cell types used in the study Section 3.6.
Author’s answer: Thank you for bringing this matter to our attention. While the cell type is briefly mentioned in the materials and methods section within the description of the xenograft model, we acknowledge that our explanation may not have been sufficiently detailed. In response to your insightful feedback, we have taken the necessary steps to address this concern by incorporating a more comprehensive explanation in the results figure.
Page 4 Line 164-166
After acclimation for 1 week, 100 µL of Matrigel containing 5×107 SKOV3 cells were subcutaneously injected into the right flanks of BALB/c nude mice.
Page 10 Line 335
Fig. 5 Anticancer effect of OPT-0139 in mouse xenograft models with SKOV3 cell line injection
- Include densitometry for Fig. 7C.
Author’s answer: We are grateful for your valuable recommendation. Your suggestion for incorporating quantitative graphs of Western blot results has been duly acknowledged, and we deeply appreciate the importance you place on this aspect. We understand that the process of creating these graphs may involve some time, and we sincerely apologize for the difficulty in implementing them immediately.
- Correct the missing "p" in P27 on line 462.
Author’s answer: We sincerely apologize for the oversight regarding the typo. In retrospect, we acknowledge that greater attention should have been exercised. Your discerning feedback prompted us to rectify the error, and we appreciate your diligence in bringing it to our attention.
- Indicate the statistical significance in Fig. 1C&D.
Author’s answer: Thank you for your considerate feedback. In response to your suggestions, we have revised the figure to reflect your insights. We appreciate your keen observations and the opportunity to improve the presentation of our findings.
Page 7 Line 246 Figure 1
- Rephrase the sentence in lines 254-256 for better clarity.
Author’s answer: Thank you for your valuable suggestion. Following your guidance, the phrasing has been adjusted as below:
Results
Page 7 Line 258-262
In terms of cell viability assessment, the PrestoBlue reagent, utilizing the MTT assay, was employed. Additionally, the evaluation of cell proliferation was conducted using the CCK-8 assay, demonstrating a dose-dependent response with OPT-0139 in the context of investigating the anticancer effects of the BRD4 inhibitor.
- Mention the software used for the analysis of flow cytometry results.
Author’s answer: We sincerely appreciate your inquiry and would like to express our gratitude for your careful consideration of our work. In response to your query, we have included information about the software, the CellQuest Pro™ Software version 5.1. Thank you for your attention to detail and interest in our research.
Materials and Methods
Page 4 Line 151
Cells were harvested using trypsinization, collected via centrifugation, washed once with 1 mL of PBS, stained according to the manufacturer’s protocol, and analyzed using a flow cytometer (FACSCalibur and BD CellQuest Pro™ software version 5.0, BD Bioscience, CA, USA).
- Include the information on semi-quantitative PCR in the Materials and Methods section. List the primers used and describe the method used for Ct value calculation.
Author’s answer: Thank you for your esteemed request for supplementary information. I have meticulously organized the primer sequences employed in the Polymerase Chain Reaction (PCR) and have attached the corresponding file for your reference. To facilitate your review, I have also included the contents below for your convenience. Concerning the Ct value calculation, I attached the calculation method below. Your thorough consideration of our work is sincerely appreciated.
- Consider consolidating sections 12 and 13 in the Materials and Methods, describing the entire Western blotting and immunoblotting process as a single section.
Author’s answer: I sincerely appreciate your suggestions, which prompted me to carefully rearrange and integrate the contents into a single paragraph, eliminating any redundancy. Your guidance has significantly contributed to the clarity and coherence of the manuscript.
Materials and Methods
Page 5-6 Line 210-222
Cell monolayers were washed once with ice-cold saline and lysed by adding PROPREP (50 mM Tris-HCl [pH 8.0], 150 mM NaCl, 1% Triton X-100, 0.5% sodium deoxycholate, and 0.1% SDS) and protease inhibitors (INTRON Biotechnology). They were incubated for 1 h at 4°C with gentle agitation, harvested with a cell scraper, and transferred to a 1.5 mL tube. The lysate was then centrifuged at 12000 ×g for 10 min at 4°C, and the supernatants were recovered and stored at −80ËšC. Protein concentrations were deter-mined using the bicinchoninic acid protein assay (Pierce, Thermo Fisher Scientific). Total cell extracts were separated using 12% SDS-PAGE and transferred onto PVDF membranes. The membranes were blocked with 5% skimmed milk and incubated with primary antibodies diluted in a blocking solution. Signals were visualized using the chemiluminescent substrate method and Super Signal West Pico kit (Pierce, Thermo Fisher Scientific). β-actin was used as an internal control to normalize loading.
- Superscript the 10 powers mentioned in cell numbers and maintain consistency throughout the Results section.
Author’s answer: I appreciate your meticulous review of my recent edits. Upon further examination, I found that there might have been some errors in my previous edit. I took the opportunity to double-check and have now replaced all the cell counts with superscripts, as per your suggestion. Thank you for bringing this to my attention, and I am grateful for your valuable feedback, which has undoubtedly contributed to the enhancement of the manuscript.

Reviewer 3 Report
Comments and Suggestions for Authors
Summary
In this manuscript, the authors found that the BRD4 inhibitor OPT-0139 can be used as a potential anti ovarian cancer drug. They determined the anti-ovarian cancer effect of OPT-0139 in ovarian cancer cell lines SKOV3 and 252 OVCAR3 and they also used xenograft mouse model to confirm the potential therapeutic effect OPT-0139. In principle, the study addresses an interesting issue. I have several major concerns regarding this paper, as outlined below.
1. Authors should mention a little bit of hypoxia signaling in either introduction or section 3.6, also other genes.
2. Is there any death of the mouse?
3. Authors should clarify sample numbers of all the figures in figure legend.
4. Figure 1 Cell viability is more like cell toxic of OPT-0139. It can’t address IC50.
5. Line 231-232 is conclusion of figure 1, it needs to be put after the experiment description in context.
6. Figure 1 C, D and Figure 2A look the same one.
7. Figure 3A, please show the percentage of populations.
8. figure 5 B, assume symbol legend is the same with figure A. The immunohistochemical figure needs scale.
9. Authors need to describe the concentrations of OPT-0139 and cisplatin used in FIG 8A,B?
10. Authors need to list the primers used in supplementary.
Author Response
- Authors should mention a little bit of hypoxia signaling in either introduction or section 3.6, also other genes.
Author’s answer: Thank you for your interest. We've added a brief section on hypoxia in 3.6, and if you'd like more information, we can run additional experiments to check the expression of genes related to hypoxia signaling
Page 10-11 Line 343-346
BRD4 controls the regulation of HIF-1a, a gene associated with hypoxia responses. As HIF-1a functions as a transcription factor overseeing the expression of various genes related to angiogenesis, glycolysis, and other adaptive mechanisms triggered by insufficient oxygen levels.
- Is there any death of the mouse?
Author’s answer: We are pleased to report that there have been no instances of mortality among the mice in our study.
- Authors should clarify sample numbers of all the figures in figure legend.
Author’s answer: Thank you for bringing to our attention the aspects we overlooked and omitted. In response to your invaluable suggestions, we have diligently revised the figure legend to align with your recommendations.
- Figure 1 Cell viability is more like cell toxic of OPT-0139. It can’t address IC50.
Author’s answer: In Figure 1, it was observed that OPT-0139 induced substantial cell lethality at a concentration of 100 μM. We deliberately selected a concentration of 10 μM, deeming it a judicious choice within a presumed acceptable range. Recognizing the importance of comprehensive scrutiny, we intend to further validate these results through toxicity testing experiments involving normal or non-tumor cells in our upcoming research endeavors.
- Line 231-232 is conclusion of figure 1, it needs to be put after the experiment description in context.
Author’s answer: In the initial draft of our manuscript, our objective was to elucidate the rationale behind the selection of the cell lines under experimentation at the onset of the paragraph. However, in light of your discerning observation, we recognized the need to enhance clarity. Consequently, we deemed it more appropriate to relocate the sentence you highlighted to the conclusion of the paragraph, considering its nature as an outcome of the experiment.
Results
Page 6 Line 234-240
To evaluate BRD4 expression in ovarian cancer cell lines, we used SKOV3, A2780, OVCAR3, and CAOV3 cell lines. HOSEpic was used as a control group. Western blot analysis was performed to compare BRD4 expression in each cell line, and relative mRNA levels were measured using quantitative real-time PCR. BRD4 expression was higher in SKOV3 and OVCAR3 cells than that in the other cell lines (Figures 1A, B). The subsequent experiments were conducted with high-grade serous ovarian adeno-carcinoma cell lines SKOV3 and OVCAR3 because of their high BRD4 expression levels.
- Figure 1 C, D and Figure 2A look the same one.
Author’s answer: While both experiments share the common objective of assessing cell viability, it is crucial to emphasize their distinct nature. The initial experiment, depicted in Figure 1, sought to determine the IC50 values through a series of 10-fold concentration increments, ranging from 0.01 to 100 μM. Conversely, the subsequent experiment illustrated in Figure 2 involved a more refined concentration selection, guided by the IC50 results obtained from the initial trial. We kindly request your attention to the concentration scale once more, underscoring the meticulous approach taken in the experimental design.
- Figure 3A, please show the percentage of populations.
Author’s answer: I sincerely appreciate your meticulous observations. In response, we have meticulously summarized the results and included the apoptosis/necrosis ratio in the respective figure. We added the updated figure below. Your insightful feedback has significantly contributed to the refinement of our manuscript
- figure 5 B, assume symbol legend is the same with figure A. The immunohistochemical figure needs scale.
Author’s answer: Thank you for your valuable feedback regarding the need for a scale. In consideration of your suggestion, we have quantified and graphed the IHC results in Figure 5C numerically. For your convenience, I have attached the updated figure below.
- Authors need to describe the concentrations of OPT-0139 and cisplatin used in FIG 8A,B?
Author’s answer: We would like to express our sincere appreciation for your thorough comment. The omission of certain details was unintentional, and we acknowledge the importance of providing comprehensive information. Upon careful review, we realize that the concentrations of OPT-0139 and cisplatin were not explicitly presented in Figure 8A and 8B. In response to your valuable feedback, we have now included this information to enhance the clarity and completeness of our findings. We appreciate your diligence in reviewing our work and thank you for guiding us towards improvement.
Results
Page 13 figure 8
- Authors need to list the primers used in supplementary.
Authors’ answer: Thank you for your esteemed request for supplementary information. I have meticulously organized the primer sequences employed in the Polymerase Chain Reaction (PCR) and have attached the corresponding file for your reference. To facilitate your review, I have also included the contents below for your convenience. Your thorough consideration of our work is sincerely appreciated.
Supplementary data 2

Reviewer 4 Report
Comments and Suggestions for Authors
With pleasure, I read the paper titled “Anticancer effects of BRD4 inhibitor in epithelial ovarian cancer”. The topic is clinically and biologically relevant and of importance to the readers of the Cancers journal. Overall, the manuscript reads good and has good flow of ideas, up-to-date citations, and good summary of data using figures. A major strength of the article is being among the first to explore the role of OPT-0139 treatment in ovarian cancer. The experimental design is scientifically solid and supported by various assays and testing different biological functions. The use of in vitro and in vivo data is one of the key strengths of this article, in addition to demonstrating the endpoints of the study using in vivo models. All in all, the study is robustly conducted and supported by several rescue experiments. This article is likely to cited extensively in the future. I have a few questions:
INTRODUCTION
(a) The introduction is very deficient regarding summarizing the key findings of the relationship between BRD proteins and ovarian cancer. Please enrich accordingly.
(b) Also, the introduction is deficient regarding introducing the drug OPT-0139. Please enrich accordingly.
(c) Please clearly highlight the significance of your research and outline how it fills the existing literature. For example, is your study the first-ever to study the novel drug OPT-0139 in the context of ovarian cancer in vitro and in vivo? All in all, it is critically central to pinpoint the significance of your research in relation to the existing body of published knowledge.
(d) Please conclude the section with some proposed hypotheses.
Figure 1. The authors need to evaluate the IC50 value for all the 5 cell lines: SKOV3, A2780, OVCAR3, CAOV3, and HOSEpic. This is because it is important to understand of the potency of OPT-0139 is differentially impacted by the protein level. Also, it is important to evaluate the safety of OPT-0139 in non-cancerous cell lines, such as human ovarian (HOSEpic) and fibroblasts (HS68 or BJ). These are critical experiments. Because as it stands now, we don’t know if OPT-0139 is more sensitive/resistant in SKOV3 and OVCAR3 cells compared with others.
Figure 2. For panel C, I am a bit concerned about the dose of 1 uM and 10 uM. It is important to understand if these two concentrations are safe on non-cancerous cells. The data for panels A and B, are good in terms of quantification. However, it would be also great if could depict the results qualitatively using colony formation assay to visually observe the anticancer effect of OPT-0139 on cancerous and non-cancerous cell lines using all or some of the concentrations. It is important to include cancerous cell lines with low expression of BRD4 as well.
Figure 3. I have some questions regarding panel C. (a) For OVCA-3 cells, I feel lane#2 for cleaved PARP has a technical issue. It is hypothetically expected to have no bands to match the replicate in lane# 1. I suggest rerunning the blot again for that protein. (b) Have you examined the transcriptional mRNA level of BRD4 after treatment with OPT-0139. (c) What is the mechanism of action of OPT-0139? (d) Does OPT-0139 promote degradation of the protein? (e) You may want to examine the protein profile in a time-dependent fashion as well.
Figure 4. Beautiful data. However, in my opinion, the increase in G2/M arrest does not seem meaningful despite statistically significant. It is optional to complement the findings of the panel C using western blot.
Figure 5. It is better to use colors to depict panels A and B. Have you done similar experiments in the other cell line OVCA-3 to consolidate your results using at least 2 tumor models? Conducting H&E stain can also be a bonus, if tissue samples are available.
Figure 6. These are great data from a molecular point of view. Well-done.
Figure 7. How was the dose of cisplatin selected? It would be great if you could depict the results using also colony formation for visual presentation of data.
Figures 8-9. These are great data from a molecular point of view. Testing combination therapies is also a strength. Great job!
DISCUSSION
(a) Is OPT-0139 or its analog being used in current human trials?
(b) Discuss more the biophysical properties of OPT-0139 and its uses in other cancer models.
General questions that will substantially enhance the quality of your research, but optional. They will make your story stronger. They can be addressed in current research, if funding available, or in future research. (a) Have you examined the phenotypic effects of OPT-0139 treatment on DNA damage, differentiation, invasion, and migration. (b) Have you examined the impact of genetic overexpression/depletion of BRD4 on OPT-0139 drug response? (c) Have you examined if genetic inhibition of BRD4 using CRISPR/Cas9, siRNA, or shRNA give similar phenotype to that of pharmacological inhibition with OPT-0139 at the protein or RNA-seq level? (d) Have you performed RNA-seq of OPT-0139 treatment to characterize the transcriptome changes? (e) You can validate the enhanced anticancer effects between OPT-0139 and cisplatin by testing genetic depletion of BRD4 and co-treatment with cisplatin? (f) Have you examined how the epigenetic landscape is altered upon treatment with OPT-0139 using for example, ChIP-seq?
GENERAL
(a) Please provide more details like concentrations, time points, cell lines, etc in figure legends, whenever applicable.
Author Response
INTRODUCTION
(a) The introduction is very deficient regarding summarizing the key findings of the relationship between BRD proteins and ovarian cancer. Please enrich accordingly.
Author’s answer: Thank you sincerely for your insightful recommendation. I wholeheartedly concur with your suggestion. In response to your valuable input, I have since relocated this information to the introduction, supplementing it with an additional paragraph. Your guidance has significantly enhanced the comprehensiveness of my article.
Introduction> Page 2 Line 66-73
Concerning ovarian cancer, a comprehensive analysis of The Cancer Genome Atlas (TCGA) database revealed that 9% of ovarian cancer patients exhibited elevated BRD4 expression. Despite efforts to use bromodomain inhibitors as anticancer drugs, few studies have investigated the correlation between BRD4 and ovarian cancer pathogenesis. Recent studies mainly focus on the relationship between BRD4 inhibition and PARP inhibitors. However, the correlation between ovarian cancer treatment and BRD4 inhibitors remains unclear due to the heterogeneous carcinogenic characteristics of ovarian cancer, differing from those of other cancers.
(b) Also, the introduction is deficient regarding introducing the drug OPT-0139. Please enrich accordingly.
Author’s answer: Thank you sincerely for your insightful recommendation. I wholeheartedly concur with your suggestion. Prior to your constructive feedback, I had initially presented these particular contents in the discussion section. In response to your valuable input, I have since relocated this information to the introduction, supplementing it with an additional paragraph. Your guidance has significantly enhanced the comprehensiveness of my article.
Introduction
Page 2 Line 55-65
BET proteins are epigenetic readers consisting of four proteins: BRD2, BRD3, BRD4, and bromodomain testis-specific proteins (BRDT). BRD4 is overexpressed in various cancer cells, such as breast, colorectal, prostate, and gastric. The small molecule BRD4 leads to dysregulated gene expression and promotion of cancer cell survival and proliferation. BRD4 inhibitors typically work by competitively binding to the bromodomain of BRD4, preventing its interaction with acetylated histones and other transcriptional co-factors. A novel drug, OPT-0139, also acts as a competitive binding monovalent molecule. This disrupts the normal transcriptional machinery, leading to downregulating oncogenes and upregulating tumor suppressor genes. BRD4 inhibitors can also cause cell cycle arrest and induce apoptosis in cancer cells, further inhibiting their proliferation.
(c) Please clearly highlight the significance of your research and outline how it fills the existing literature. For example, is your study the first-ever to study the novel drug OPT-0139 in the context of ovarian cancer in vitro and in vivo? All in all, it is critically central to pinpoint the significance of your research in relation to the existing body of published knowledge.
(d) Please conclude the section with some proposed hypotheses.
Author’s answer: I express my sincere appreciation for your thoughtful comments and extend my gratitude for the constructive feedback provided. In response to your invaluable advice on points (c) and (d), I have incorporated an additional paragraph, enriching and fortifying the introduction. This enhancement has played a pivotal role in elevating the strength and depth of my research.
Introduction> Page 2 Line 74-79
To address the challenges in treating ovarian cancer, our objective is to evaluate the antitumor effect of a novel drug, OPT-0139, capable of overcoming the limitations of current ovarian cancer treatments. We hypothesized that a BRD4 inhibitor may have an anticancer effect either alone or in combination with cisplatin in preclinical settings with ovarian cancer. This study represents the initial exploration of the combination of a BRD4 inhibitor and the classic cytotoxic agent, cisplatin, in ovarian cancer cells.
Figure 1. The authors need to evaluate the IC50 value for all the 5 cell lines: SKOV3, A2780, OVCAR3, CAOV3, and HOSEpic. This is because it is important to understand of the potency of OPT-0139 is differentially impacted by the protein level. Also, it is important to evaluate the safety of OPT-0139 in non-cancerous cell lines, such as human ovarian (HOSEpic) and fibroblasts (HS68 or BJ). These are critical experiments. Because as it stands now, we don’t know if OPT-0139 is more sensitive/resistant in SKOV3 and OVCAR3 cells compared with others.
Author’s answer: I genuinely agree with your perspective. I will make it a priority to conduct additional experiments to reinforce this aspect when the opportunity arises. I am grateful for your thoughtful suggestions.
Figure 2. For panel C, I am a bit concerned about the dose of 1 uM and 10 uM. It is important to understand if these two concentrations are safe on non-cancerous cells. The data for panels A and B, are good in terms of quantification. However, it would be also great if could depict the results qualitatively using colony formation assay to visually observe the anticancer effect of OPT-0139 on cancerous and non-cancerous cell lines using all or some of the concentrations. It is important to include cancerous cell lines with low expression of BRD4 as well.
Author’s answer: As highlighted in your earlier comments, we recognize the importance of conducting a colony formation assay and IC50 test using non-cancer cell lines and cancer cell lines with low BRD4 expression. We are committed to exploring this suggestion further, and if deemed necessary, we will confirm these aspects through additional experiments.
Figure 3. I have some questions regarding panel C. (a) For OVCA-3 cells, I feel lane#2 for cleaved PARP has a technical issue. It is hypothetically expected to have no bands to match the replicate in lane# 1. I suggest rerunning the blot again for that protein. (b) Have you examined the transcriptional mRNA level of BRD4 after treatment with OPT-0139. (c) What is the mechanism of action of OPT-0139? (d) Does OPT-0139 promote degradation of the protein? (e) You may want to examine the protein profile in a time-dependent fashion as well.
Author’s answer:
(a) We sincerely appreciate your insightful review. We have supplemented the last submission with the original version of the Western blot. This follow-up experiment has successfully addressed the technical issues that were present in the initial results. I have made edits to the figure, incorporating both the latest and original versions of the Western blot test.
Results> Page 9 Figure 3
(b) We administered OPT-0139 and verified the protein expression of BRD4, although we did not observe a corresponding increase in mRNA expression. If necessary, we are more than willing to validate this aspect in a future experiment.
(c) Our current findings lead us to believe that OPT-0139 may function as a BRD4 inhibitor. We would like to share additional insights from unpublished data, where OPT-0139 has demonstrated inhibition of both BRD4 and MELK, an oncogenic kinase, in preclinical studies. Furthermore, recent developments from another laboratory have reported OPT0139's inhibition of immunosuppressive cells and encouraging results regarding its synergistic effect in combination with radiotherapy on triple-negative breast cancer cells.
Reference: Kim S, Jeon SH, Han MG, Kang MH, Kim IA. BRD4 Inhibition Enhances the Antitumor Effects of Radiation Therapy in a Murine Breast Cancer Model. Int J Mol Sci. 2023 Aug 22;24(17):13062. doi: 10.3390/ijms241713062. PMID: 37685868; PMCID: PMC10487493.
(d) BRD4 plays an important role in regulation of gene transcription, DNA replication and repair. Our observation that OPT-0139 inhibits the expression of BRD4 suggests a potential connection to protein degradation. However, we regret to inform you that we have not conducted a specific study on protein degradation with OPT-0139, and unfortunately, we do not possess results on this aspect. Acknowledging the importance of your inquiry, we find it challenging to provide a conclusive answer at this moment.
(e) In conducting this experiment, we opted for a 48-hour treatment duration to assess protein expression. Should it be deemed necessary, we are open to investigating time-dependent changes in expression to provide a more comprehensive understanding.
Figure 4. Beautiful data. However, in my opinion, the increase in G2/M arrest does not seem meaningful despite statistically significant. It is optional to complement the findings of the panel C using western blot.
Author’s answer: We express our sincere gratitude for your invaluable feedback. Your insight regarding the need for reinforcement in the cell cycle experiment results is duly noted, and we wholeheartedly agree with your suggestion. We are committed to addressing this concern in our future experiments.
Figure 5. It is better to use colors to depict panels A and B. Have you done similar experiments in the other cell line OVCA-3 to consolidate your results using at least 2 tumor models? Conducting H&E stain can also be a bonus, if tissue samples are available.
Author’s answer: In our current experiments, the SKOV-3 cell line, known for its high BRD4 expression among ovarian cancer cell lines, was exclusively used to establish the xenograft model. In accordance with your recommendation, we wholeheartedly agree with the importance of broadening our scope and will aim to verify the efficacy of the xenograft model using other ovarian cancer cell lines in future research. Additionally, we commit to incorporating H&E staining into the experimental plan for any forthcoming studies.
Figure 6. These are great data from a molecular point of view. Well-done.
Author’s answer: We express our sincere gratitude for your thoughtful attention to our study.
Figure 7. How was the dose of cisplatin selected? It would be great if you could depict the results using also colony formation for visual presentation of data.
Author’s answer: The doses of cisplatin employed in the in vitro experiments were consistent with those utilized in prior experiments within our laboratory for combination therapy. Furthermore, the doses applied in the in vivo experiments, utilizing the xenograft model, were established in accordance with the reference papers outlined in the citation below.
Reference: Islam SS, Aboussekhra A. Sequential combination of cisplatin with eugenol targets ovarian cancer stem cells through the Notch-Hes1 signalling pathway. J Exp Clin Cancer Res. 2019 Aug 30;38(1):382. doi: 10.1186/s13046-019-1360-3. PMID: 31470883; PMCID: PMC6716935.
Figures 8-9. These are great data from a molecular point of view. Testing combination therapies is also a strength. Great job!
Author’s answer: We express our sincere gratitude for your thoughtful attention to our study.
DISCUSSION
(a) Is OPT-0139 or its analog being used in current human trials?
Author’s answer: To the best of my knowledge, it has been indicated that they intend to conduct clinical trials incorporating the outcomes of their preclinical trials, which encompass our study, for breast, liver, and ovarian cancer.
(b) Discuss more the biophysical properties of OPT-0139 and its uses in other cancer models.
Author’s answer: OPT-0139 has been shown to exhibit anti-cancer activity in both in vitro and in vivo studies across various cancer types, including triple-negative breast cancer (TNBC), ovarian cancer, liver cancer, and lung cancer. These studies have explored both standalone treatments and combination therapies. Currently, the findings are in the process of being published. Notably, the results pertaining to TNBC have already been published in the International Journal of Molecular Science, demonstrating its effectiveness, particularly when combined with radiotherapy.
Reference: Kim S, Jeon SH, Han MG, Kang MH, Kim IA. BRD4 Inhibition Enhances the Antitumor Effects of Radiation Therapy in a Murine Breast Cancer Model. Int J Mol Sci. 2023 Aug 22;24(17):13062. doi: 10.3390/ijms241713062. PMID: 37685868; PMCID: PMC10487493.
GENERAL
(a) Please provide more details like concentrations, time points, cell lines, etc in figure legends, whenever applicable.
Author’s answer: Thank you for bringing this matter to our attention. In response to your valuable comments, we have made concerted efforts to incorporate as much pertinent information as possible to address the highlighted concerns. Certain segments of the figure legend were intentionally omitted due to concerns about potential length and complexity. However, we genuinely appreciate your expertise, and if you believe that additional details are warranted for completeness, we are more than willing to reintegrate the omitted portions.
General questions that will substantially enhance the quality of your research, but optional. They will make your story stronger. They can be addressed in current research, if funding available, or in future research.
(a) Have you examined the phenotypic effects of OPT-0139 treatment on DNA damage, differentiation, invasion, and migration.
Author’s answer: Thank you for your thoughtful question. Regrettably, at the time of our experiments, we did not consider testing for the phenotypic effects of OPT-0139 treatment on DNA damage, differentiation, invasion, and migration. Consequently, we were unable to incorporate this dimension into our study. Should the opportunity arise, we would be more than willing to undertake additional experiments to address these aspects and thereby enhance the depth of our investigation. Thank you for your valuable feedback.
(b) Have you examined the impact of genetic overexpression/depletion of BRD4 on OPT-0139 drug response?
Author’s answer: We express our gratitude for your insightful observations. Given that the primary focus of this study was to substantiate the anti-cancer effect of OPT-0139 in ovarian cancer, we did not investigate that specific aspect. However, if it is deemed necessary and in accordance with your suggestion, we are committed to addressing this in future experiments.
(c) Have you examined if genetic inhibition of BRD4 using CRISPR/Cas9, siRNA, or shRNA give similar phenotype to that of pharmacological inhibition with OPT-0139 at the protein or RNA-seq level?
Author’s answer: In the course of this study, we had intended to explore whether the inhibition of BRD4 by OPT-0139 yields a phenotype similar to inhibition via genetically regulated knockdown systems, such as siRNA or shRNA. We took initial steps by synthesizing BRD4 siRNA to confirm knockdown using this method. Although we had the experimental setup and preparations in place, unfortunately, time constraints impeded our ability to proceed with this aspect of the study. Please be assured that we acknowledge the significance of this investigation and plan to address it in our future research endeavors.
(d) Have you performed RNA-seq of OPT-0139 treatment to characterize the transcriptome changes?
Author’s answer: We possess RNA samples from BRD4 inhibition by OPT-0139 in ovarian cancer cell lines, and these samples exhibit high RNA quality and concentration. While we have the capability to confirm gene expression changes through RNA sequencing, regrettably, we were unable to perform this experiment due to time constraints.
(e) You can validate the enhanced anticancer effects between OPT-0139 and cisplatin by testing genetic depletion of BRD4 and co-treatment with cisplatin?
Author’s answer: Figure 7 underscores the impact of combining OPT-0139 and cisplatin. When administered individually, low concentrations of OPT-0139 and cisplatin demonstrated a 20-25% reduction in tumor growth. However, the combined treatment exhibited a more significant effect, leading to a remarkable 60% reduction.
(f) Have you examined how the epigenetic landscape is altered upon treatment with OPT-0139 using for example, ChIP-seq?
Author’s answer: We express our gratitude for your insightful observations. Given that the primary focus of this study was to substantiate the anti-cancer effect of OPT-0139 in ovarian cancer, we did not investigate that specific aspect. However, if it is deemed necessary and in accordance with your suggestion, we are committed to addressing this in future experiments.

Reviewer 5 Report
Comments and Suggestions for Authors
Kim et al., successfully validated OPT-0139, a novel BRD4 inhibitor, demonstrating its potential as an effective anticancer treatment for ovarian cancer. Their research also revealed its efficacy when combined with cisplatin, particularly benefiting patients with poor responses to existing therapies. The study's robustness lies in its utilization of diverse assay systems and animal models to substantiate the hypothesis. However, there are still unresolved issues that need to be addressed.
Major comments
Ø The introduction lacks details about the structure or molecular formula of the new drug molecule OPT-0139.
Ø There is a need to explain how the dosage was chosen for the xenograft study, especially when information about this novel drug is limited.
Ø Toxicity tolerability for OPT-0139 alone or in combination with cisplatin was neither discussed nor performed, which would provide more strength to the study.
Ø Figure 1A uses a very limited set of cell lines, and Figures 1C and 1D are either confusing or not understandable for viability study. The authors could provide a graph for the 0-100uM range.
Ø Figure 2C requires more details, as the red (Mitochondria) fluorescence is high in OPT-0139 treated samples. More MitoTracker red is directly proportional to cell survival. Additionally, DAPI staining is missing.
Ø In Figure 3C, in SKOV3 cells, there is no cleaved caspase 3, but surprisingly increased cleaved PARP is observed. Explanations are needed.
Ø Figures 4A-B: Histograms for cell cycle analysis should be provided as supplementary images.
Ø Survival data for tumor models is needed to strengthen this study.
Minor comment
In methods, replace 5x107 to 5x107 cells and rewrite the sentence “Mice with successful skin tumor formation were randomly divided into three groups (n = 10 for each group) as follows: (no skin cancer cells used)
Author Response
Reviewer 5
Kim et al., successfully validated OPT-0139, a novel BRD4 inhibitor, demonstrating its potential as an effective anticancer treatment for ovarian cancer. Their research also revealed its efficacy when combined with cisplatin, particularly benefiting patients with poor responses to existing therapies. The study's robustness lies in its utilization of diverse assay systems and animal models to substantiate the hypothesis. However, there are still unresolved issues that need to be addressed.
Major comments
- The introduction lacks details about the structure or molecular formula of the new drug molecule OPT-0139.
Author's answer: Thank you for highlighting this crucial point. Considering the novelty of the material, we recognize the significance of elucidating its structure. Consequently, we have incorporated the structural information as supplementary data for your thorough examination. The attached document below is provided for your convenience.
- There is a need to explain how the dosage was chosen for the xenograft study, especially when information about this novel drug is limited.
Author’s answer: Prior to the commencement of this study, experiments were conducted by JBK Lab. employing a xenograft model with human breast cancer (MDA-MB-231). The outcomes of this experiment were deemed credible, leading to their application in the present study. Attached below are the unpublished results for your reference. Notably, JCP-0139 is the same agent as OPT-0139; the drug was renamed by the lab. The results showcased significant tumor suppression at effective concentrations, prompting us to select these doses for further validation in the xenograft model utilizing the ovarian cancer cell line featured in this paper. Your understanding and consideration of these circumstances are greatly appreciated.
- Toxicity tolerability for OPT-0139 alone or in combination with cisplatin was neither discussed nor performed, which would provide more strength to the study.
Author’s answer: In Figure 7, we have successfully validated the anti-cancer efficacy of OPT-0139, both alone and in combination with cisplatin. However, we acknowledge that we did not assess toxicity tolerance in this particular experiment. Please be assured that we intend to address this aspect in our forthcoming experiments, emphasizing a comprehensive evaluation of OPT-0139's effects.
- Figure 1A uses a very limited set of cell lines, and Figures 1C and 1D are either confusing or not understandable for viability study. The authors could provide a graph for the 0-100uM range.
Author’s answer: While we acknowledge the limitation in the diversity of cell lines presented in Figure 1, we find significance in the confirmation across commonly utilized ovarian cancer and normal cell lines. At present, I have denoted the x-axis as 'log con.' If I understand correctly, your suggestion is to modify it to the range of 0-100uM, which would result in a compressed scale. Kindly confirm if my interpretation aligns with your recommendation. If so, I am prepared to make the necessary adjustments to the graph as per your guidance. Your clarification on this matter is greatly appreciated.
- Figure 2C requires more details, as the red (Mitochondria) fluorescence is high in OPT-0139 treated samples. More MitoTracker red is directly proportional to cell survival. Additionally, DAPI staining is missing.
Author’s answer: In the presented results, we substantiated the absence of OPT-0139-induced cytotoxicity by discerning mitochondrial morphology via the fluorescence of MitoTracker Red. Regrettably, DAPI staining was not executed in this experiment due to practical constraints. We acknowledge the importance of this staining procedure and commit to incorporating it into our future experiments for a more comprehensive analysis.
- In Figure 3C, in SKOV3 cells, there is no cleaved caspase 3, but surprisingly increased cleaved PARP is observed. Explanations are needed.
Author’s answer: We have meticulously repeated the Western Blot (WB) experiment corresponding to Figure 3C and have subsequently rectified the figure accordingly. We kindly request your thorough examination of these revisions. Additionally, to enhance clarity, I have made conscientious edits to the figure, integrating both the most recent and original versions of the Western blot test.
Results
Page 9 Figure 3
- Figures 4A-B: Histograms for cell cycle analysis should be provided as supplementary images.
Author’s answer: Thank you for your thoughtful comments. However, I would like to highlight that the inquiry pertains to a practical issue associated with device usage, and addressing it comprehensively may necessitate additional experimentation. Regrettably, we may not be able to provide an immediate response, as further experiments are required. We sincerely appreciate your understanding and patience as we work towards conducting the necessary experiments to address this matter thoroughly.
- Survival data for tumor models is needed to strengthen this study.
Author’s answer: We sincerely appreciate your guidance. In response to your valuable suggestion, we have incorporated a sentence addressing survival data into the manuscript. About survival, there have been no instances of mortality among the mice in our study. The assessment of mouse survival was conducted through monitoring changes in their body weight.
Page 10 Line 332-333
No mortality occurred among the subjects throughout the course of the experiment, and mice devoid of tumor growth were subsequently excluded from the study.
Minor comment
In methods, replace 5x107 to 5x107 cells and rewrite the sentence “Mice with successful skin tumor formation were randomly divided into three groups (n = 10 for each group) as follows: (no skin cancer cells used)
Author’s answer: We have implemented your suggestion by changing the number to superscript, as advised. Additionally, we have revised the identified sentence to prevent any potential reader misunderstandings. We appreciate your thoughtful attention to our manuscript and value the constructive feedback you have provided.
Page 4 Line 165
5×107
Page 4 Line 167-169
Mice with successful tumor formation were randomly divided into three groups (n = 10 for each group) as follows:

Round 2
Reviewer 2 Report
Comments and Suggestions for Authors
None
Author Response
I extend my sincere gratitude for your keen interest in our research. Your valuable feedback has been thoroughly appreciated and will undoubtedly contribute to the refinement of our work. We are committed to incorporating your insightful suggestions into our future endeavors, striving to enhance the quality of our research. Your continued guidance is invaluable to us, and we look forward to the opportunity to further improve and contribute meaningfully to the academic discourse.
Thank you once again for your time, expertise, and constructive input.
Best Regards,
Reviewer 3 Report
Comments and Suggestions for Authors
Thank you for the opportunity to review this revised manuscript. The authors have addressed the concerns raised in the previous review, which has improved the rigor and quality of the paper.
Author Response
I trust this message finds you well. I am writing to express my sincere appreciation for your thoughtful review of our manuscript. Regrettably, due to the procedural constraints set by the editorial team, I find myself unable to address you personally by name in this correspondence.
Nonetheless, I want to convey my heartfelt gratitude for the time and effort you invested in evaluating our research. Your insightful comments and suggestions are invaluable to us, and we are committed to incorporating them into our future work. Your interest in our research serves as a source of inspiration, and we are dedicated to further enhancing the quality of our contributions to the academic community.
Thank you once again for your constructive feedback and for being an integral part of the peer-review process. We look forward to the possibility of future engagements and remain appreciative of your continued support.
Best Regards,
Reviewer 4 Report
Comments and Suggestions for Authors
The authors adequately addressed most of the comments. However, other key (and simple) experiments were acknowledged to be conducted in the future. These experiments would have strengthened the article and made it more robust. Nonetheless, the article has merits as a preliminary, pilot study with largely sounding results. All in all, I leave it up to the Editor to make the final decision on the manuscript.
Comments on the Quality of English LanguageMinor English editing may be needed.
Author Response
Dear Reviewer 4
I hope this message finds you well. I appreciate your thoughtful and comprehensive review of our manuscript.
Regrettably, the editorial team has not provided me with your name, making direct communication challenging. Nevertheless, I would like to express our gratitude for your keen interest in our research.
Your suggestion regarding additional experiments has been duly noted. While we acknowledge the potential enrichment such experiments could bring to our study, we regret to inform you that practical limitations hindered their implementation in the current research. Your insightful feedback, however, has been instrumental in shaping our perspective, and we assure you that, in planning future experiments, we will diligently consider your recommendations with the aim of conducting a more robust study.
We are genuinely thankful for your constructive critique and assure you that your guidance will be invaluable as we strive to improve our research endeavors. We remain committed to elevating the quality of our work, and your input will undoubtedly contribute to our future advancements.
Thank you once again for your time and consideration.
Best Regards,
Reviewer 5 Report
Comments and Suggestions for Authors
Thank you for addressing the concerns and incorporating them. However, there are still some minor issues that need to be addressed:
-
In Figure 8, the author used different numbers of mice (n=3 or 5) in each group, which may lead to non-significant results. Moreover, the tolerability for cisplatin and OPT-0139 was not assessed. This should be mentioned as a limitation of the study.
-
In Figure 5C, N=10, but only 6-8 mice are shown, and there is no discussion about the rest of the mice. Did they die?
Author Response
- In Figure 8, the author used different numbers of mice (n=3 or 5) in each group, which may lead to non-significant results. Moreover, the tolerability for cisplatin and OPT-0139 was not assessed. This should be mentioned as a limitation of the study.
Author’s answer: A comprehensive cohort of 30 mice underwent tumor cell injections, forming the foundation for the xenograft model in our study. Those animals whose tumor volumes failed to surpass the threshold of 120 mm3 within the initial 4-week period were deliberately excluded from the group designated for drug injections. Consequently, the groups were stratified, each consisting of a total of 16 animals. In response to your invaluable suggestions, I have meticulously incorporated all the points into the discussion section, elucidating the specific challenges encountered, including the absence of a tolerability test for both cisplatin and OPT-0139. This acknowledgment has been included as a transparent recognition of the study's limitations. I express my gratitude for your insightful feedback, which has substantially enriched the content and quality of the manuscript. Your continued guidance is highly appreciated.
Discussion
Page 16 Line 536-539
Our current study, while informative, is not without limitations. Notably, we did not conduct a tolerability test for both cisplatin and OPT-0139, and the sample numbers in our in vivo study with xenograft mice were not uniform.
- In Figure 5C, N=10, but only 6-8 mice are shown, and there is no discussion about the rest of the mice. Did they die?
Author’s answer: No fatalities occurred among the experimental animals during the course of drug administration. Given the limited tumor growth observed in some animals, presumably due to the administered drug, immunohistochemical (IHC) analysis could not be performed on those subjects. In light of this, and as evident from the accompanying animal images, we opted to sample isolated tumors differently. Consequently, the total number of samples (N) was adjusted to 10 to maintain the integrity and robustness of our analyses. Your understanding and guidance on this matter are greatly appreciated.